# Data-driven design of shape-programmable magnetic soft materials

Alp C. Karacakol [1,2], Yunus Alapan [1,3,4] ✉, Sinan O. Demir [1,5] & Metin Sitti [1,5,6] ✉

Magnetically responsive soft materials with spatially-encoded magnetic and material properties enable versatile shape morphing for applications ranging from soft medical robots to biointerfaces. Although high-resolution encoding of 3D magnetic and material properties create a vast design space, their intrinsic coupling makes trial-and-error based design exploration infeasible. Here, we introduce a data-driven strategy that uses stochastic design alterations guided by a predictive neural network, combined with cost-efficient simulations, to optimize distributed magnetization profile and morphology of magnetic soft materials for desired shape-morphing and robotic behaviors. Our approach uncovers non-intuitive 2D designs that morph into complex 2D/3D structures and optimizes morphological behaviors, such as maximizing rotation or minimizing volume. We further demonstrate enhanced jumping performance over an intuitive reference design and showcase fabrication- and scale-agnostic, inherently 3D, multi-material soft structures for robotic tasks including traversing and jumping. This generic, data-driven framework enables efficient exploration of design space of stimuli-responsive soft materials, providing functional shape morphing and behavior for the next generation of soft robots and devices.

Stimuli-responsive materials, which respond to external stimuli such as light, pH, and magnetic or electrical fields[1–4], have gained significant attention for enabling complex shape-morphing in untethered structures[5–7], and shown distinct advantages in fields like object manipulation, soft robotics, wearable devices, and biomedical applications[1,2,8–10]. Among various responsive composites, magnetic soft materials stand out due to their rapid advancements in micron-scale resolution, three-dimensional (3D) directionality, multi-material compositions, and complex 3D structures[11–16]. These capabilities enable a wide range of static and dynamic shape-morphing behaviors across different length scales, from micro to milli.

Magnetic soft materials consist of hard magnetic particles (e.g., $Nd_2Fe_{14}B$, $CrO_2$) embedded within a soft material matrix (e.g., silicone rubber, polydimethylsiloxane (PDMS), hydrogels) and spatially

magnetized at desired orientations to create a distributed magnetic moment. When subjected to an external magnetic field, the particles experience torques that align their magnetization with the field, generating programmable shape deformations. This spatial programmability is achieved through techniques such as jig-assisted assembly[11], lithography[12,15], 3D printing[13], and local heat-assisted magnetization in micromachined structures[14]. These methods allow precise control over material composition, magnetization orientation and magnitude, and structural features, enabling complex shape-morphing capabilities.

The capability to finely tune and spatially program these different materials and structural properties in magnetic soft materials enable a vast design space for complex shape morphing. However, the majority of the literature relies on intuition-based trial and error (Edisonian) approaches despite the highly non-intuitive-design space, arising from

[1]Physical Intelligence Department, Max Planck Institute for Intelligent Systems, Stuttgart, Germany. [2]Department of Mechanical Engineering, Carnegie Mellon University, Pittsburgh, PA, USA. [3]Department of Mechanical Engineering, University of Wisconsin-Madison, Madison, WI, USA. [4]Department of Biomedical Engineering, University of Wisconsin-Madison, Madison, WI, USA. [5]Stuttgart Center for Simulation Science, University of Stuttgart, Stuttgart, Germany. [6]School of Medicine and College of Engineering, Koç University, Istanbul, Turkey. ✉e-mail: alapan@wisc.edu; sitti@is.mpg.de

the inherent coupling between the magnetic and mechanical responses. In a unit segment, the ratio of magnetic and soft materials affects the magnetization strength along with elastic modulus. Similarly, the cross-sectional area and thickness of a segment (i.e., morphology/structural configuration) determine magnetic strength and bending stiffness, which are inversely related[17]. Magnetic torque on the segment also nonlinearly changes during deformation depending on the misalignment between its magnetization and the external magnetic field direction[18]. Furthermore, different magnetization directions in neighboring segments could result in incomplete alignment with the external field, due to the counteracting mechanical forces arising in the continuous segments. Overall, the mechanical continuum and distributed magnetic profile along with these coupling effects govern the relative deformation and position of each segment throughout the structure[4]. In such multi-dimensionally coupled systems, a marginal change in magnetic programming or morphology of a local segment could significantly change the shape-morphing of the whole structure (Fig. S1). Non-intuitive shape-morphing, stemming from such intricate coupling effects, renders Edisonian approaches infeasible to explore the vast design space enabled by magnetic soft materials. The previous approaches for design optimization of magnetic soft materials were mostly limited to pre-defined simple morphologies (e.g., rods, beams), two-dimensional (2D) magnetic profiles, and 2D planar deformations[11,19–24]. Efficient design strategies for the spatial programming of morphology (structural optimization) and 3D magnetization profile in magnetic soft materials with 3D structure and multi-material composition for desired 2D and 3D shape-morphing or behavior are yet to be shown.

Here, we present a data-driven design approach to spatially program both 2D/3D morphology, 3D magnetic profile, and multi-material composition of magnetic soft materials for desired 3D shape-morphing and behavior. The design approach relies on continuous exploration of the design space through randomly generated candidates and exploitation of promising designs via stochastic variations guided by a predictive neural network (NN) model. The selected promising designs are tested in a computationally cost-effective simulation engine capable of evaluating the dynamic behaviors of magnetic soft materials. The resultant best-performing designs for desired 2D and 3D shape-morphing of beams inspired by mathematical functions and intricate sharp-cornered shapes are experimentally demonstrated, showcasing the *sim2real* transfer. The developed design strategy is also employed to achieve desired dynamic behaviors, without any defined target shape-morphing, including maximizing turn number, maximizing height, and minimizing bounding sphere volume for magnetic soft beams and sheets. The superiority of the data-driven approach is further highlighted in the high-performance jumping behavior of magnetic soft robots, where the intuitive design adapted from the literature[25] failed to lift from the surface. The proposed framework is also utilized to design magnetic soft materials with 3D structure and multi-material composition for configurable robotic behaviors of traversing, and vertical and directional jumping, highlighting its fabrication, programming, and applicability across scales. The data-driven design strategy introduced here provides an efficient and versatile platform, capitalizing on the extensive design space enabled by the advances in fabrication and programming capabilities, thus unlocking the potential of stimuli-responsive soft materials toward real-world applications.

## Results

### Data-driven design framework

The data-driven design strategy aims to achieve desired quasi-static and dynamic shape-morphing behaviors for magnetic soft materials by optimizing the spatial programming of the magnetic profile and morphology with respect to pre-defined external magnetic fields (Fig. 1A–C). Our magnetic soft materials are composed of ferromagnetic neodymium-iron-boron ($Nd_2Fe_{14}B$) microparticles (5–25 μm diameter) dispersed in an elastomer. For 2D structures, magnetic soft elastomer sheets are laser micromachined into desired morphologies and heat-assisted magnetic programming[14] is employed for spatial magnetic encoding (Fig. S2A–C), which results in distributed 3D magnetization directions over the morphology (Fig. 1B). Multi-material magnetic soft 2D/3D structures are built by voxel-based assembly of individual pre-magnetized voxels of different materials (Fig. S3). The resolution of morphological features and distributed magnetization are represented by voxel and segment parametrization, respectively (Fig. 1B). The distributed magnetization segment matrix, defined as magnetic profile (**M**), is represented directly in spherical coordinates ($\mathbf{M}_\theta$, $\mathbf{M}_\varphi$). Morphology is determined by mapping the spatial coordinates of voxels to material type (e.g., 1 for magnetic material and 0 for empty voxels), utilizing a compositional pattern-producing network (CPPN)[26], as CPPNs have shown to be beneficial in reducing the parameter number without the loss of generalizability for soft material morphology representation[27].

The exploration of the coupled vast design space resulting from co-optimization of the morphology and 3D magnetic profile is enabled by the developed data-driven design algorithm and a computationally low-cost simulation engine (Fig. 1D, E). A heuristic design space exploration method of multi-dimensional archive of phenotypic elites (MAP-elites)[28] is guided by a surrogate model of a neural network (NN), establishing an interconnected data-driven algorithm (Figs. 1D and S4A, B, and SI S1). The performances of generated design candidates are predicted within the simulation environment (Fig. 1D). The developed parallelizable simulation environment utilizes a mass-spring lattice model involving both translational and rotational springs[29] coupled with magnetic forces and torques, and completes a dynamic simulation within a few minutes (Fig. 1E and SI S2). Evaluated candidates establish a design repertoire within a user-chosen low-dimensional feature space (e.g., filled voxel ratio, net magnetic moment), illuminating the effect of these features over the design space (Fig. 1F). The process of design generation and performance prediction is repeated until the defined threshold of performance or maximum generation number is reached. The final best-performing design is experimentally realized, displaying sim2real transfer, as demonstrated for a triangular wave shape-morphing under magnetic field (Fig. 1G and Supplementary Movie S1). The algorithm benchmarking is also established by comparing our algorithm with original MAP-elites, random search and other MAP-elites variants. Comprehensive benchmarking for various performance metrics including QD-score, global performance, reliability, precision, coverage, and performance over iterations highlights the effectiveness of our algorithm for the design of stimuli-responsive soft materials (Fig. S5 and SI S1.3).

### 2D and 3D shape-morphing magnetic soft beams

To test the performance of our data-driven design framework for shape-morphing, magnetization distribution and morphology for unit beams (12 mm length × 1 mm width × 0.2 mm thickness) were determined for a variety of nonengineered intricate shapes originating from mathematical functions that are hard to design based on intuition. The design algorithm balances external magnetic field-induced local deformations against gravitational effects in beams fixed at one end. We first demonstrated a non-periodical and varying amplitude sinusoidal shape-morphing, closely matching the desired shape, in terms of period and amplitude, both in simulation and experiment (Fig. 2A and Supplementary Movie S1). We further validated the non-intuitive nature of the coupled design space by introducing marginal changes in magnetization and morphology of the sinusoidal shape-morphing beam, as well as the external magnetic field, which resulted in dramatic shifts from the desired shape in the simulation environment (Figs. S6–S8). These sensitivity analysis results show that the design space is inherently non-linear with numerous local minima and there

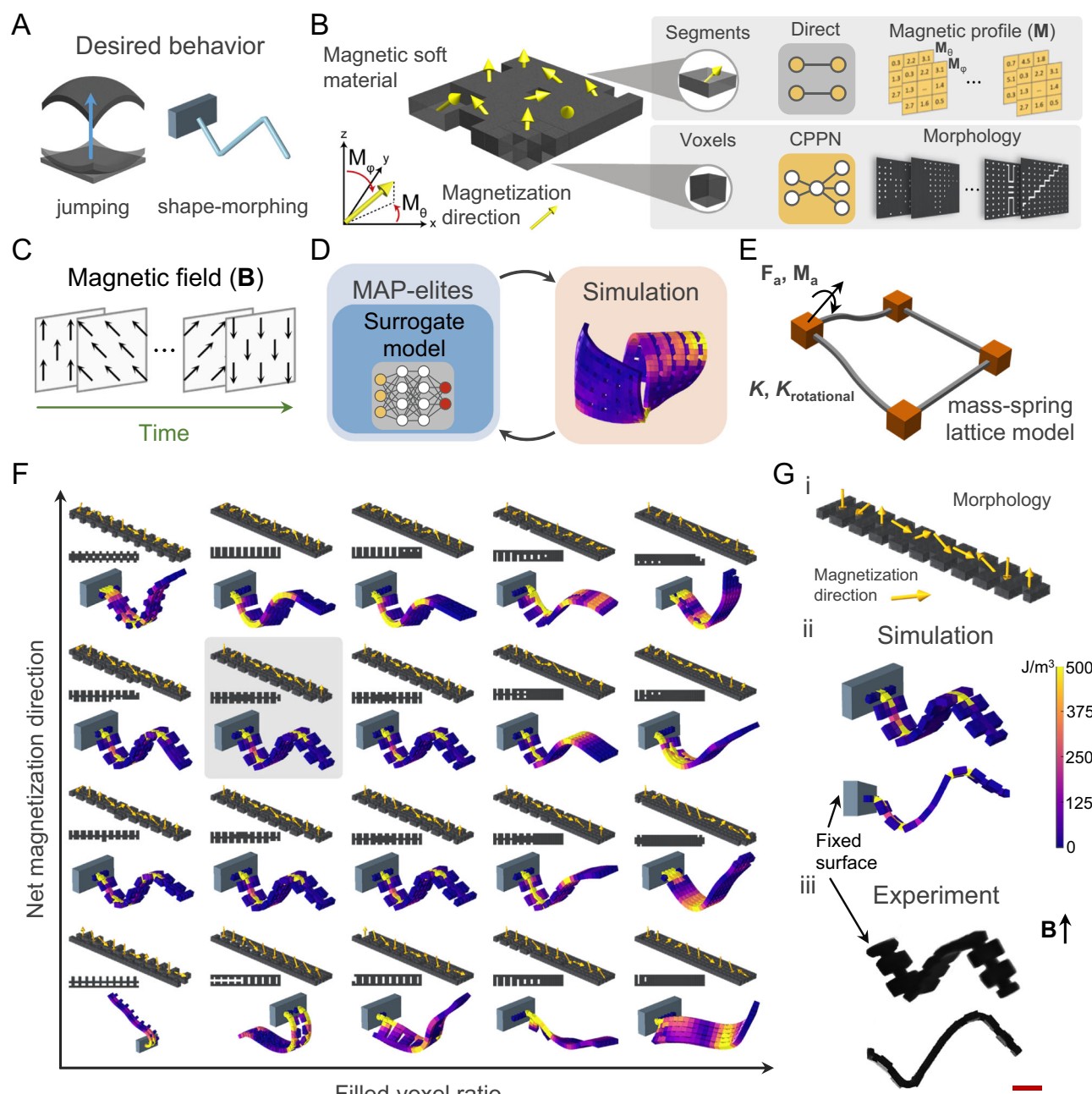

**Fig. 1 | Data-driven design of magnetically programmed soft materials.** A desired behavior, such as jumping and shape-morphing, of a magnetic soft material (**A**) is encoded by programming the morphology and distributed 3D magnetization (yellow arrows) (**B**) for a pre-defined external magnetic field over time (**C**). The magnetic soft material is parametrized into voxels and segments representing the resolution of morphological feature and magnetization, respectively. The magnetic profile (**M**) is defined in spherical coordinates ($M_\theta$, $M_\varphi$) via direct representation of distributed magnetization segment matrix. Morphology is determined by a compositional pattern-producing network (CPPN) that maps the spatial coordinates of voxels to material type. **D** The data-driven design strategy is enabled by a heuristic design space exploration method (Multi-dimensional archive of phenotypic elites, MAP-elites) guided by a neural network (NN) operating as a surrogate model.

**E** Proposed design candidates are evaluated in a custom-built, computationally low-cost simulation environment utilizing mass-spring lattice model with translational and rotational springs coupled with magnetic forces and torques. **F** For a given behavior, the data-driven design strategy generates a repertoire within a low-dimensional space for user-chosen features, such as filled voxel ratio (Feature 1) and net magnetization (Feature 2). The best design is highlighted with the gray shade. **G** Experimental realization of the best-performing design; (i) morphology and magnetic profile, (ii) predicted, and (iii) experimental shape-morphing behavior. The scale bar is 2 mm. Magnetic field (**B**) is pre-defined as 30 mT in the direction indicated by the black arrow. Magnetization directions are depicted via the yellow arrows.

are a wide range of equilibrium states for a given design, which depends on the initial design state (Fig. S7) and the induced magnetic field input sequences (Fig. S8) to the system.

While shape deformations with smooth curvatures have been abundantly shown and used in the shape-morphing materials[11,19,20,30,31], sharp deformations in soft structures are constrained by the high strain energies localized at the bending sections as can be seen in our simulation results. Our data-driven design approach addresses this challenge by balancing magnetic torques, bending stiffness, and gravity, as shown in examples of a step signal (Fig. S9A and Supplementary Movie S1) and a square signal (Fig. 2B and Supplementary Movie S1) with 2 and 4 sharp corners, respectively. Another challenge

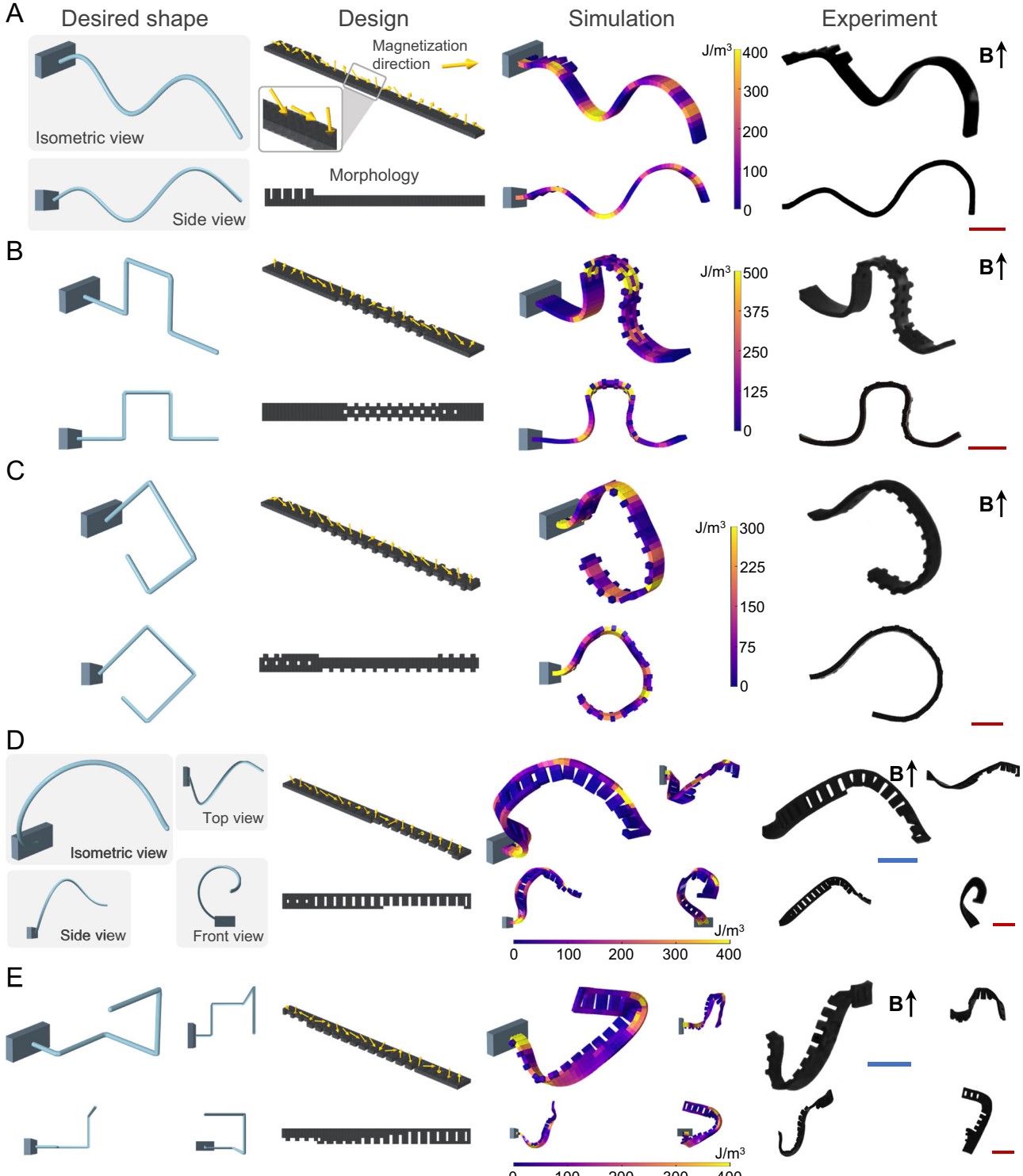

**Fig. 2 | Data-driven design of morphology and magnetic profile for 2D and 3D shape-morphing magnetic soft beams.** The desired shape-morphing from different views depicted as averaged lateral voxel positions on the longitudinal axis, data-driven best-performing designs with morphology and magnetic profile, predicted shape-morphing, and experimental shape morphing under an external magnetic field for 2D varying sinusoidal signal (**A**), square signal (**B**), diamond (**C**) shapes, and 3D spherical spiral (**D**) and step (**E**) shapes. Design spaces for 2D and 3D shape-morphing are calculated as ~2.7e141, and ~3.4e186, respectively. Scale bars, 2 mm. Actuation is performed by applying magnetic field (**B**) of 30 mT in the direction indicated by the black arrow. Magnetization directions are depicted via yellow arrows. Color bars indicate the average strain energy density.

unique to magnetic soft materials is creating tip deformations over 180 degrees from the initial orientation under constant magnetic fields since the final position and orientation of the tip are highly coupled to the deformation of the rest of the structure. We initially showed the potential of our approach in designing such tip deformations through a Fibonacci-inspired shape-morphing with a total tip rotation of 270 degrees (Fig. S9B and Supplementary Movie S1). Next, we also demonstrated a diamond shape-morphing structure, combining

4 sharp corners with a tip rotation of 225 degrees (Fig. 2C and Supplementary Movie S1). Overall, intricacy of our shape-morphing demonstrations, quantified by a shape-complexity score (SI S3), is 2–20 times greater than previous examples in the literature[11,19,20,32] (Fig. S11 and Table S1).

Our data-driven design strategy can be further expanded to 3D shape-morphing of beams, which could necessitate the incorporation of bending and twisting around all axes rather than bending around a single axis as in 2D deformations, rendering intuition-based designs significantly more challenging. Beams morphing into 3D helices of a single period with constant (Fig. S9C and Supplementary Movie S2) or varying amplitude (Fig. 2D and Supplementary Movie S2) were realized under constant magnetic fields without environmental constraints. We further designed beam structures shape morphing in 3D with multiple sharp corners (Figs. S9D and 2E, and Supplementary Movie S2), inspired by the evolved antenna structures[33]. We observed that our data-driven designs overcame the challenging deformation angles by bending and twisting simultaneously, as evident in the localized high strain energies around deforming sections. Also, slight deviations in simulation results for some of the demonstrations (Table S2) could be ascribed to the physically unrealizable nature of the desired shapes, as these were chosen without prior knowledge of feasibility. Nevertheless, performance evolution graphs show initial rapid improvements, followed by a slower but steady increase in performance for both 2D and 3D shape-morphing cases, which could alleviate these deviations with extended algorithm runs (Fig. S10). The demonstrated intricate 2D and 3D shape-morphing of soft beam structures underline the capability of the developed approach to systematically explore the coupled morphology and magnetic profiles, resulting in extensive design spaces of around 2.7e141 and 3.4e186, respectively (Table S3).

## Design of magnetic soft structures for morphological tasks

Shape-morphing in magnetic soft materials can be utilized to generate a functional behavior, ranging from locomotion to gripping[31,34]. However, conventionally, functional behaviors are intuitively defined through several temporal shape-morphing states, such as open and closed forms of a gripper. Subsequent design of the magnetization profile and morphology of the magnetic soft materials, along with the external magnetic field control, are further defined intuitively to generate the desired shape-morphing states. Overall, such existing intuitive designs achieve functional behavior through either cycling between different shape-morphing states[15] or changing the direction of the external magnetic field while conserving the shape-morphing state, including surface rolling locomotion and corkscrew motion-induced swimming[35,36]. Despite abundant examples of functional behaviors shown in the literature, they are based on discovered magnetization and morphological designs, along with external fields, rather than systematically programmed for the desired functionality.

The data-driven design strategy described here enables direct programming of desired functional behaviors through magnetization and morphology optimization of magnetic soft structures without any restrictions to biased shape definitions as utilized in previous intuitive approaches. The capability of our data-driven strategy to design for functional behaviors was initially demonstrated through a series of morphological tasks. First, a fixed-end beam was designed to maximize the turn number around the longitudinal axis under a constant magnetic field (Fig. S12A and Supplementary Movie S3). While the magnetic soft beam was able to complete a full turn, consistent with the simulation result, its position was shifted downwards, which is plausible given the fact that the design algorithm solely aims for the rotation around the longitudinal axis of the beam without trying to balance the gravity or other effects. We further extended our design strategy to optimize the magnetization and morphology of a free-form magnetic soft material sheet ($10 \times 10 \times 0.2$ mm) to maximize the height of its center point (Fig. S12B and Supplementary Movie S3). Data-driven

algorithm-generated design was able to achieve a height of 3.7 mm (0.37 body length) for its center point under a constant magnetic field of 30 mT. Similarly, an optimized design was generated for minimizing the bounding sphere volume of a magnetic soft sheet ($6 \times 6 \times 0.2$ mm), achieving a minimum bounding sphere radius of 3.09 mm, resulting in a volume of 123.6 mm³ with a 2.6-fold volume reduction from 319.3 mm³ (Fig. 3A and Supplementary Movie S3). These demonstrations show the capability of our data-driven strategy to design magnetic soft structures to achieve desired behavioral tasks, in which the design space can reach up to ~5.4e1233 (Table S3).

## Jumping behavior for magnetic soft millirobots

An interesting functional behavior for magnetic soft millirobots, other than locomotion and object manipulation, is jumping, which requires a sudden release of energy to overcome the gravity. Most conventional soft materials with relatively slow actuation require the utilization of bistable mechanism designs for generating sudden movements[37,38]. On the other hand, magnetic soft materials with rapid response times can achieve jumping by gaining enough momentum through fast interaction with surfaces[25]. However, the momentum gained inherently depends on spatial interaction with the surface, interaction speed, and stored elastic energy, all of which are directly determined by the material properties (e.g., remanent magnetization strength, elastic modulus), magnetization profile, and morphology of the magnetic soft robots.

We employed our data-driven design strategy to develop a small-scale magnetic soft millirobot with high-performance jumping. To quantitatively compare the performance of our data-driven design, we adapted an existing intuitive design and external magnetic field signal from the literature[25], with the caveat of 3-times smaller magnetization strength of our magnetic material (due to $Nd_2Fe_{14}B$ particles with a lower Curie temperature for heat-assisted magnetic programming as different from the commonly used ones in the literature with much higher Curie temperature and magnetization strength) (Table S4). When both our data-driven design and the intuitive design adapted from the literature are programmed into a magnetic soft millirobot ($3.6 \times 1.4 \times 0.2$ mm) using the same fabrication and programming method and actuated under reversing magnetic fields of 10 mT, the intuitive design failed to lift from surface at all both in simulation and experiment (Figs. 3B, C and S13A). On the other hand, the magnetic soft robot with the data-driven design (generated from a design space of around 1.8e53, Table S3) achieved a jumping height of ~1.4 mm, equivalent to 0.39 body length, under the same conditions (Fig. 3D and Supplementary Movie S4).

It was previously claimed that the jumping performance of magnetic soft robots could significantly deteriorate when the width-to-length aspect ratios are closer to 1 with the intuitive magnetization profile and morphology[25]. For a sheet-shaped magnetic soft robot with an aspect ratio of 1, we employed our data-driven framework to optimize the 3D magnetization profile and morphology for enhanced jumping behavior (Fig. S13B and Supplementary Movie S4). Expansion in the morphology workspace ($6 \times 6$ mm) and the 3D magnetic profile significantly amplifies the design space from around 1.8e53 (for the beam structure) to around 1.2e752 (Table S3). In contrast to the intuitive design employed in the literature, the data-driven algorithm-based design generated a jumping height of around 1.9 mm equivalent to 0.43 body length, achieving a slightly better jumping performance than the aforementioned beam-shaped soft robots. These results show that discovered intuitive magnetization profile designs cannot be transferred to different morphologies and materials, which require systematic approaches for task-specific designs, as exemplified in our design strategy.

## Design of multi-material and 3D magnetic soft millirobots

Magnetic soft materials have recently been combined with other stimuli-responsive and passive materials with varying elastic modulus

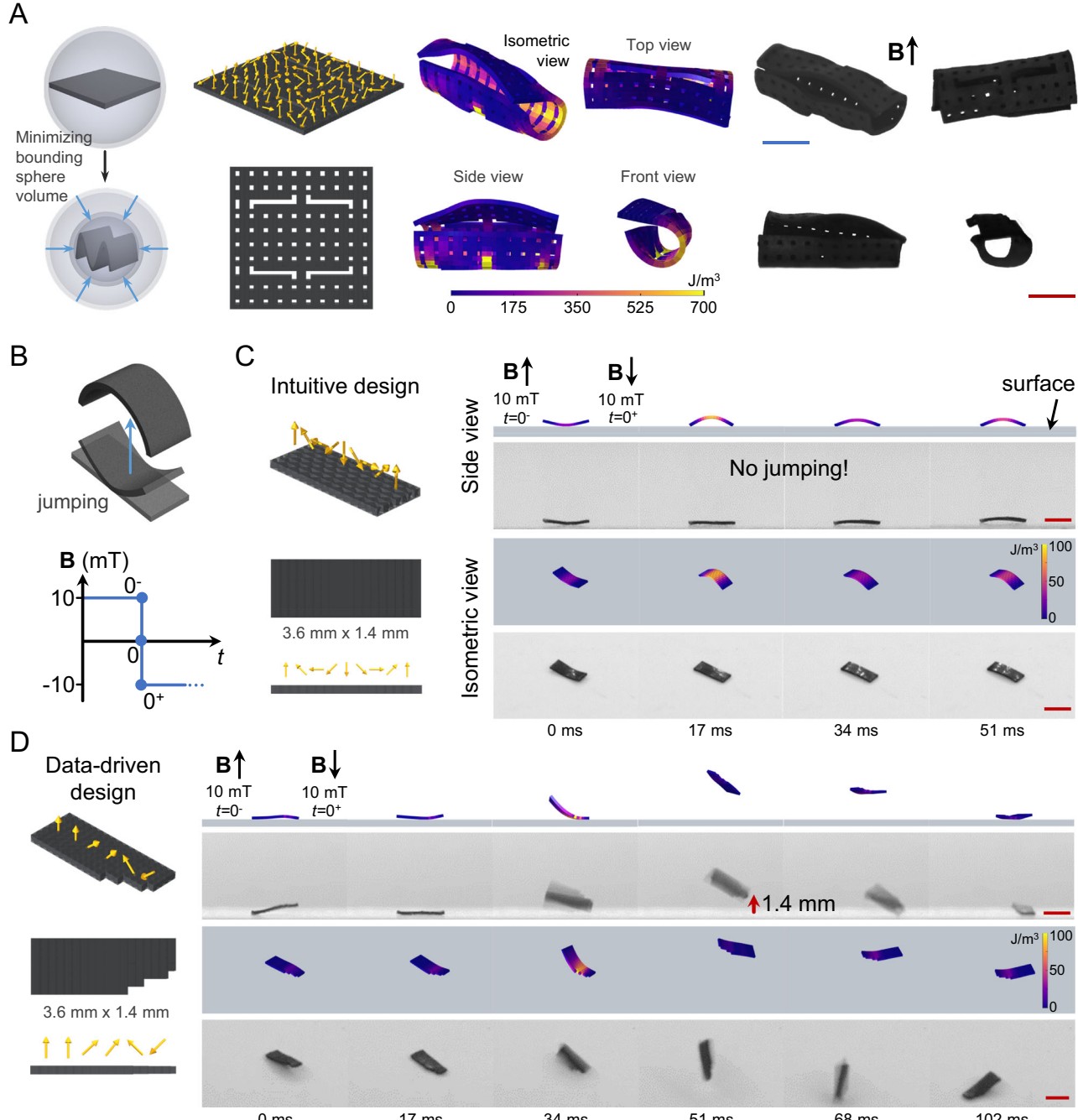

**Fig. 3 | Data-driven design of desired behaviors in magnetic soft structures and millirobots. A** Conceptual drawing of the desired morphological task of minimizing the bounding sphere volume of a sheet, data-driven best-performing design with morphology and magnetic profile, predicted behavior, experimental realization under external magnetic field of 30 mT in the direction indicated by the black arrow. **B** Conceptual drawing of a beam-shaped magnetic soft millirobot jumping via a sudden change in the magnetic field direction. Intuitive magnetic soft millirobot design adapted from the literature[25] with a continuous magnetic profile and a uniform beam morphology (**C**) and data-driven magnetic soft robot design with a discrete magnetic profile incorporating arbitrary changes of magnetization in adjacent segments and non-uniform morphology (**D**). Dynamic behaviors of the intuitive and data-driven designs in simulations and experiments, as shown in side (top rows) and isometric (bottom rows) views. The intuitive design failed to generate any jumping behavior, defined by the minimum distance of the magnetic soft robot to the surface at peak, as also predicted by the computational simulations. On the other hand, the data-driven design achieved -1.4 mm, 0.39 body length, jumping height. Actuation is performed by applying a uniform magnetic field (**B**) of 10 mT in the direction indicated by the black arrows. Design spaces are -1.2e752 and -1.8e53 for the sheet structure and beam robots, respectively. Scale bars, 2 mm. Magnetization directions are depicted via yellow arrows. Color bars indicate the average strain energy density.

to achieve shape-morphing structures and robots with multi-functionality, environmental adaptation, and enhanced robustness[16,39–41]. While advances in fabrication techniques allow spatially encoding material type and response in 2D and 3D structures, designing material profiles along with spatial programming of stimuli response is a daunting task, given the enormous design space due to the wide variety in material selection and response. In addition, the scale of desired structures or robots, that could range from micrometers to centimeters, would require different fabrication techniques with critical implications for the design process. Therefore, the design

strategies should be compatible with different fabrication techniques and capable of handling the enormous design space resulting from material selection, spatial stimuli response, and 3D structural complexity at varying scales. The data-driven strategy proposed here can also address the design challenge of soft materials with different fabrication methods, multi-material composition, and 3D structures due to its generic and versatile framework (Fig. 4A). To demonstrate the compatibility of our approach in with other fabrication methods, we adapted voxel-based assembly[16] capable of fabricating 3D structures with multi-material compositions into our framework (Fig. S3). We created a material palette consisting of passive and magnetically responsive materials with soft and rigid variations (Figs. 4B and S14A, and Table S4), expanding the design space by a power of 2.5 for any given structural workspace. The magnetization directions of the magnetic materials are discretized into six discrete primary Cartesian axes directions to simplify the fabrication process (Figs. 4B and S14B). The flexible nature of the proposed design strategy allows facile adaptation of the constraints for the parameter representation arising from the requirements of different fabrication methods, as well as material and structural complexity.

We first employed our strategy in designing a multi-material 2D structure, with a workspace of $9 \times 5 \times 1$ voxels, for vertical jumping, resulting in a design space of ~2.9e66 (Table S4). The best-performing design (Fig. S14C) achieved a jumping height of ~5 mm (0.28 body length) experimentally with a comparable behavior to simulation results (Fig. S15, and Supplementary Movie 5), showing the capability of our approach in designing 2D structures with multi-material composition for robotic behaviors. Next, we extended our design strategy to intrinsically 3D structures with multi-material compositions for a diverse range of robotic behaviors, including vertical and directional jumping, as well as traversing locomotion (Fig. 4C–E, and Supplementary Movie 5). The workspace for these demonstrations was set to $7 \times 7 \times 5$ voxels with a design space of ~7.8e361 (Table S3). The best-performing designs (Fig. S14D, E) for vertical and directional jumping demonstrated a jumping height and distance of ~5 mm (0.36 body length), and ~20 mm (1.43 body length), respectively, under the same reversing magnetic fields of 10 mT (Fig. 4C, D and Supplementary Movie 5). The data-driven design for vertical jumping structure heavily relied on soft variants of passive and magnetic materials, whereas the directional jumping structure included both soft and rigid counterparts. These demonstrations highlight the extensive design possibilities for distinct behaviors under the same control input by spatially encoding material composition and response. In addition, we showed the data-driven design of a 3D multi-material structure for traversing locomotion (Figs. 4E and S14F, and Supplementary Movie 5) under quarterly rotating and cycling magnetic fields, generating ~6.5 mm (0.46 body length) displacement per cycle. This demonstration further shows prowess of our data-driven strategy in designing 3D multi-material structures for repeating motion under continuously changing control signals. Additionally, the incorporation of well-established control strategies, including the utilization of net magnetization moment for rolling locomotion[13,15,16,25] and visual feedback for programmed trajectory[42–44], can enable multimodal locomotion modes for the untethered demonstrations (Fig. S16). Other than locomotion, we further applied our framework to design multi-material structural configuration for maximizing force generation under an external field in the simulation environment (Fig. S17).

Incorporation of multiple materials responsive to different stimuli further expands the design flexibility and provides an opportunity in designing configurable robotic behaviors. We show the applicability of our strategy to design soft robots with configurable behaviors by designing 3D multi-material structures composed of thermo-responsive and magneto-thermo-responsive materials, that are rigid in room temperatures and become softer after heated (Fig. S18A). The 3D and multi-material composition of the design was optimized in the simulation environment for the desired robotic behaviors of traversing and directional jumping under the same magnetic field actuation with thermally-triggered behavior change (Fig. S18B). The best-performing design generated by our framework (Fig. S18C) showed ~2.1 mm (0.1 body length) traversing per cycle at room temperature (Fig. S18D, G and Supplementary Movie S6) and ~12.4 mm (0.62 body length) directional jumping distance after heating (Fig. S18E, G and Supplementary Movie S6), respectively, under the same magnetic actuation signal. Furthermore, the same design demonstrated vertical jumping after heating when the applied magnetic field strength is doubled, indicating the distinctiveness of designs generated by our data-driven strategy for desired behaviors under pre-defined control inputs (Fig. S18F, G). The experience-free design of these soft structures for diverse and configurable robotic behaviors not only illustrates the applicability of our data-driven strategy beyond 2D structures with single material composition, but also shows its fabrication, programming, and scale-agnostic nature. Overall, these results highlight the potential of our data-driven design strategy in developing physically intelligent small-scale magnetic soft robots[3] for functional behaviors to be employed in long-anticipated real-world applications, ranging from targeted delivery of therapeutics to minimally invasive operations.

## Discussion

Shape-morphing is utilized extensively in nature by biological organisms ranging from single cells (e.g., amoeba) to large animals (e.g., octopus) enabling physical adaptation to unstructured and constrained environments, different tasks, and physical damage. Inspired by nature, the shape reconfigurability and reprogrammability are highly desired for engineering applications requiring multi-functional operation in unstructured environments, such as medical operations in human body and wearable and haptic bio-interfacing[30,45,46]. Advances in spatial programmability of stimuli-responsive materials at high resolution has led to the development of advanced shape-morphing capabilities rivaling their counterparts in nature[1–3]. Among different responsive materials, magnetic soft materials are especially attractive for applications in enclosed, constrained, and unstructured environments due to their fast and reversible response and the safe tissue penetration of magnetic fields. Despite such advantages along with the developments in spatial programming and fabrication of magnetic soft materials, coupled nature of the magnetic and morphological properties renders the design process highly non-intuitive, thus cumbersome. The data-driven design methodology introduced here addresses the design problem of non-intuitive coupled morphology and magnetic profile configuration and establishes an efficient and experience-free way to achieve shape-morphing for desired functional behaviors of magnetic soft materials with 3D structure and multi-material composition.

The design capability of our data-driven strategy was highlighted in magnetic soft beam structures morphing into 3D complex shapes, which were difficult to achieve with intuitive-design approaches. Such complex shape-morphing of soft beam structures could be especially desirable in potential future catheter applications to achieve access to hard-to-reach regions or operations requiring complex shapes at the site of action[30]. We further showed optimization of the magnetic and morphological profile of magnetic soft structures for desired behaviors, such as minimizing the bounding volume, or given tasks, such as jumping the highest. When we tested the intuitive jumping designs reported in the previous literature with the weaker magnetic materials employed in our study, the intuitive design failed to lift from the surface. On the other hand, our data-driven design approach was able to generate jumping behavior, thus highlighting the potential of our design strategy in improving the performance of desired behaviors. The enhanced performance achieved using our framework could further enable other stimuli-responsive soft materials suffering in terms of performance (i.e., slow actuation, inferior mechanical properties)

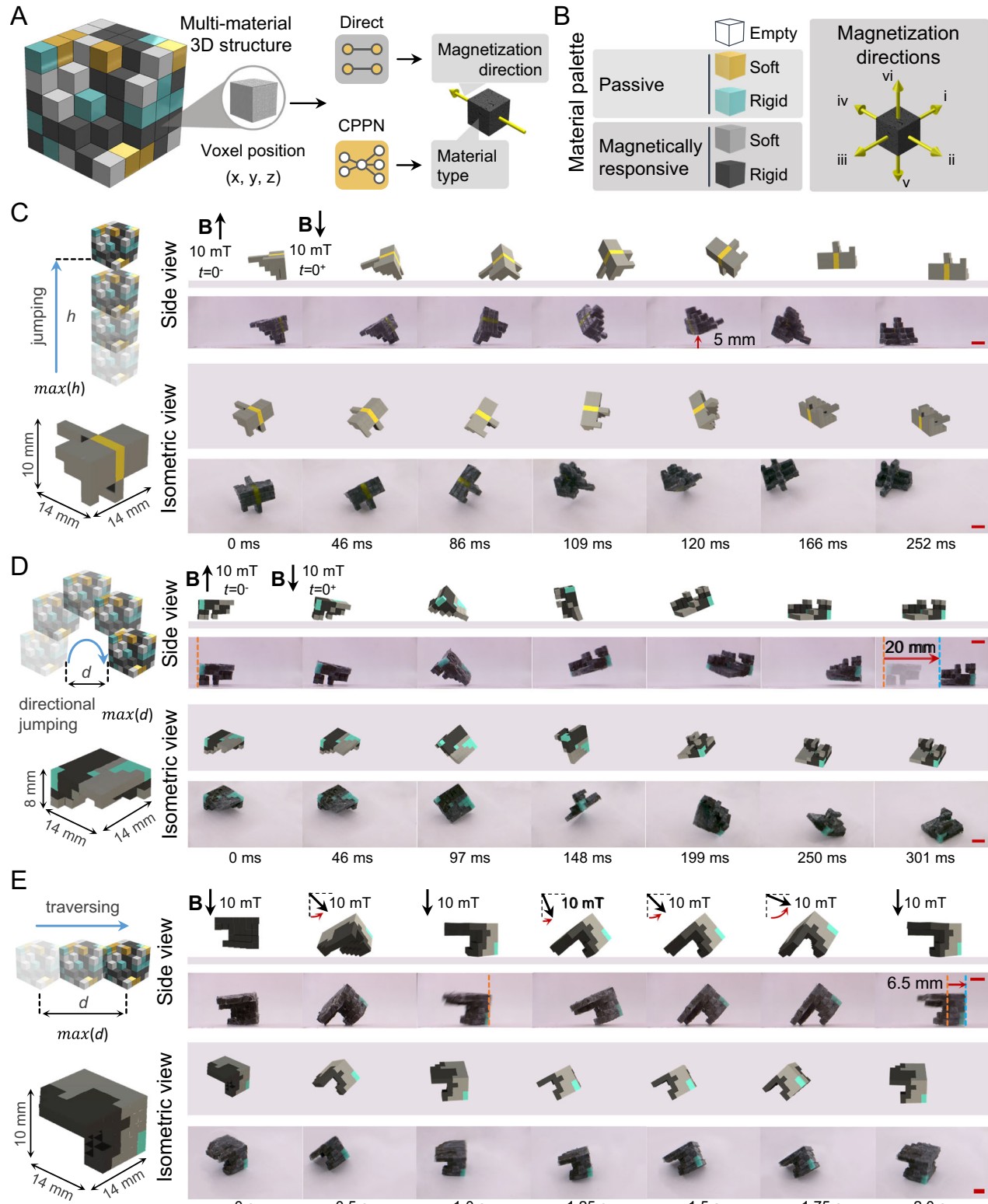

**Fig. 4 | Multi-material and 3D structural design of magnetic soft millirobots.**
**A** 3D and multi-material structural design is defined by a compositional pattern-producing network (CPPN) by mapping the voxel positions to material type. The magnetization directions are defined via direct representation of distributed magnetization voxel tensor. **B** The material palette consists of five types, including a cut out "empty" voxel, two passive and two magnetically responsive materials with soft and rigid variations. The magnetization directions of the magnetically responsive materials, depicted via yellow arrows, are restricted to six discrete primary cartesian axes directions to simplify the fabrication. The conceptual drawing of the desired robotic behavior, best-performing structural designs, and predicted behavior and experimental realization shown in side (top rows) and isometric (bottom rows) views for vertical (**C**) and directional (**D**) jumping, as well as traversing (**E**) 3D multi-material soft robots, achieving ~5 mm (0.36 body length) jumping height, ~20 mm (1.43 body length) directional jumping distance and ~6.5 mm (0.46 body length) traversing per cycle, respectively. Design space is calculated as ~7.8e361. Orange and blue dashed lines represent initial and final reference position, respectively. Scale bars, 5 mm. Actuation is performed by applying a uniform magnetic field (**B**) of 10 mT in the direction indicated by black arrows. Voxel colors indicate the material types.

but are desirable in other aspects, including sustainability[47] and biocompatibility[48]. For example, although biocompatible and biodegradable magnetic particles are highly sought after for medical applications[48], their weak magnetization strength and intuitive magnetic and morphological designs result in unsatisfactory performances for desired behaviors, which could be alleviated by the introduced data-driven approach.

Multi-material structures with desired arrangement of stimuli-responsive and passive materials in 3D provide much greater design flexibility and wider range of material functionality and robotic applications, despite the tremendous increase in design space. The generic structure of our framework, allowed us to design 3D structures with multi-material compositions, featuring materials responsive to different stimuli and with varying mechanical properties. The designed 3D and multi-material structures were built by voxel-based assembly of individual units with relatively larger volumes compared to 2D demonstrations, showcasing the scale, fabrication, and programming independent operation of our approach. Our data-driven design approach was also able to encode configurable robotic behaviors to 3D multi-material soft structures by spatially programming multiple material types responsive to different stimuli, including magnetic and temperature. While the developed framework currently optimizes the design for user-defined external stimuli, control of the external fields could be further included in the data-driven design process, providing additional design flexibility for complex dynamic behavior, temporally-resolved multi-tasking, and adaptability.

Earlier works attempting to design magnetic soft materials utilized genetic algorithms along with conventional finite element simulations, resulting in unreliable and computationally time-intensive processes, thus limiting the designs to 2D planar deformation with simplified 2D magnetic profiles and pre-defined morphologies of rods and beams[11,19,20,32] (Fig. S19A and SI S4.1). Furthermore, the comparison with other methods for the design of stimuli-responsive materials shows the latent potential of our approach, especially its capacity to deal with extremely large design spaces, which has been the bottleneck for experience-free design[27,39,49–51] (Fig. S19B and SI S4.2). While our data-driven strategy enables the design of stimuli-responsive soft materials for 3D complex shape-morphing behaviors by optimizing the 3D magnetic profile and 2D/3D morphology with multi-material compositions, the performance of the design process can be enhanced by improvements in the algorithm and the simulation environment. Although we represented the magnetic profile directly and the morphology via CPPN, their individual or coupled representation using other methods, such as variational autoencoders, and Gaussian mixture models could significantly enhance the performance in terms of the design quality and the algorithm run-time[52]. While our heuristic exploration algorithm is guided by an NN, acting as a surrogate model, its architecture is comparatively simple compared to NNs utilized in the machine learning field. NNs with higher architectural complexity could enable the development of advanced surrogate models for efficient prediction of the shape-morphing of magnetic soft materials, as well as other stimuli-responsive soft materials. The 5 million designs generated in this work, made available through an open-access database, could enable the testing, development, and adaptation of a myriad of NNs for the design of magnetic soft materials. Furthermore, the demonstrated performances could be enhanced by increasing the maximum iteration number, as observable by the improvements up to the pre-determined iteration limits (Fig. S13C, D). On the other hand, advancements in the simulation engine in terms of computation time, model precision, and contact modeling for environment interaction would significantly improve the sim2real gap and reliable synthetic data generation for the design of magnetic soft materials.

## Methods

### Simulation environment

A mass-spring lattice model coupled with magnetic forces and torques is used as a simulation engine for the dynamic behavior of magnetic soft materials. A version of the "Voxelyze" environment[29] is adapted for mechanical deformations and modified for enabling magnetic materials. "Voxelyze" implements a mass-spring lattice model with translational and rotational springs and is capable of efficiently simulating heterogeneous 3D soft bodies dynamically. Magnetic forces and torques, calculated according to a pre-defined permanent magnet used in the experiments, are integrated into the simulation platform. The values in Table S4 are used for density, Young's modulus ($E$), and magnetic remanence ($M_r$). The Poisson's ratio is assumed to be 0.49. The respective voxel sizes of the demonstrations can be found in Table S5. The computationally low-cost predictions realized in the developed simulation environment enabled the exploration of the presented design space.

Validation of the developed simulation engine is performed through a fixed-end beam magnetized along its longitudinal axis via vibrating-sample magnetometer (VSM) under a uniform magnetic field of 1.6 T. The magnetic field for actuation is generated vertically with an electromagnetic coil setup providing uniform fields. Experiments were conducted for seven different beam samples (12 mm length × 1.6 mm width × 0.2 mm thickness) under uniform magnetic field strength of 0, 0.5, 1, 2, 3, 5, 7, 10, and 13 mT. Simulation results under the same conditions for experimentally measured values of density 2.41 g/cm³, Young's modulus ($E$) 200 kPa, and magnetization of 28.6 kA/m show a similar trend to the experiments (Fig. S20A). The accuracy of the simulation engine is further improved by the fitting parameters of $E$ and magnetic remanence. Fitted values of $E$ 150 kPa and magnetic remanence of 20 kA/m are found after the fitting process utilizing Bayesian optimization (BO) (Fig. S20A and SI S2). For the jumping demonstration, damping parameters that are affecting the dynamic behavior are fitted by using the experimental results of vertical jumping presented in ref. 25. Fitted internal, collision, and global damping parameters are set to 1, 0.01, and 0.001, respectively.

The computation time of the simulation engine for magnetic soft materials is characterized for various voxel numbers and voxel size ranges (Fig. S20B–D). The computation time of a single run of 0.5-s dynamic simulation takes on average ~86 wall clock seconds for demonstrated beams (12 mm length × 1 mm width × 0.2 mm thickness) with 300 voxels (200 μm) on an Intel E5-1650 v4 processor. As the whole simulation engine is implemented in platform-agnostic C++, multiple simulation runs are easily parallelizable.

### Parameter space representation

Magnetic soft material morphology is parametrized into voxels, which are the smallest geometrical feature size. Representation of the morphology is determined by a compositional pattern-producing network (CPPN). The CPPN maps the spatial coordinates of the voxels in the workspace to the type of voxel. In this work, 0 for cut voxels (empty) and 1 for magnetic soft material are used. CPPNs for each design candidate are initialized with 5 nodes and 5 connection links. Activation functions utilized by CPPNs are "sin(x)", "±abs(x)", "±square(x)", "±sqrt(abs(x))", "square(sin(x))+x", "Gaussian", periodic "triangle wave", "square wave", "rectified sine wave", and traditional NN functions of "relu", "elu", "tanh", "swish".

The magnetic profile (**M**), defined in spherical coordinates of the magnetic soft material, is divided into segments depicting the defined magnetization resolution size for the magnetic programming method. Direct representation is used for showing **M** resulting in distributed magnetization segment matrices ($\mathbf{M}_\theta$, $\mathbf{M}_\varphi$), mapping the segments to **M**. More details on parameter space representation can be found in SI S1.1.

## Data-driven design algorithm

Multi-dimensional archive of phenotypic elites (MAP-elites) and a neural network (NN) are coupled to construct the data-driven algorithm. MAP-elites is used as a heuristic design space exploration method, and a NN is operated as a surrogate model guiding the exploration. During the evaluation of the design candidates in the developed simulation engine, NN is trained in parallel. NN is used as a pseudo model to predict the performance quality of the generated candidates to have an educated guess on promising candidates. Then, the most promising candidate population of 50 is selected for evaluation within the simulation engine. This process is repeated until the defined threshold, the number of evaluations, is reached. Detailed algorithm flow and pseudo-code can be seen in Fig. S4A, B.

MAP-elites code is adapted and modified from ref. 53. Filled voxel ratio over the workspace and net magnetization directions of ($M_\theta$ and $M_\varphi$) in spherical coordinates are chosen as the features of the MAP-elites. For 2D magnetic profiles, only $M_\theta$ is used. Mutation operations are done randomly either on morphology or magnetic profile. Mutation on morphology adds/removes a node, adds/removes an edge, or changes the weight of an existing edge on CPPN, while magnetic profile mutations apply a randomly sampled Gaussian mutation on a randomly selected segment (Fig. S4C). Crossover operation is defined as the interchange between the morphology and magnetic profile. Randomly chosen design candidates exchange their magnetic profiles ($M_\theta$ and $M_\varphi$ parameter matrices) (Fig. S4D).

NN is constructed with "Keras" as dense layers with a dropout ratio of 0.1. The input layer is structured as the total node number of morphology voxels and magnetic profile segments, followed by 6 dense layers with 128 nodes using "tanh" as an activation function. The final layer of 1 node for regression is added to these layers with the "sigmoid" activation function. The Adam optimizer is utilized for training with a default learning rate of 0.001. The loss is defined as a mean-squared error, and the batch size is set to 512. More details on the algorithm can be found in SI S1.2.

The algorithm is run on a cluster environment, and each iteration on average takes around 110 s in which the simulation makes up about 85% of the time, resulting in around 166 h of wall clock time for 5000 iterations, resulting in a total evaluations number of 2.5e5 designs for a population of 50 on 50-cores processors in parallel (SI S1.4 and Table S6).

## Fabrication and programming of magnetic 2D soft structures

Magnetic soft materials are prepared by mixing $Nd_2Fe_{14}B$ (MQFP-10-8.5HD-20180, Magnequench, Neomaterials, Toronto, Canada) microparticles into Ecoflex 00-30 silicone rubber (Smooth-on, Macungie, PA, USA) at a 2:1 ($Nd_2Fe_{14}B$: Ecoflex) mass ratio. For this, $Nd_2Fe_{14}B$ particles are mixed with Ecoflex-30 A and mixed thoroughly for 3 min. Then, part B of Ecoflex-30 is added to the mixture and further mixed for another 3 min. The final mixture is degassed under vacuum for 5 min to eliminate any entrapped air. Next, the mixture is cast into molds composed of 200 μm thick tapes adhered to a flat PMMA substrate, and cured at room temperature for 4 h (Fig. S2A). Magnetic soft devices are cut into the desired morphology with an ultraviolet laser system (LPKF ProtoLaser U3, Garbsen, Germany) (Fig. S2B).

An optical profilometer (VK-X250, Keyence, Osaka, Japan) is used to measure the thickness of the magnetic soft materials, and the cut size accuracy of the laser-machining for square-shaped cuts with 1 mm, 0.8 mm, 0.6 mm, 0.4 mm, and 0.2 mm edge length (Fig. S21J). To characterize the Young's modulus ($E$) and the strain at the break of the magnetic soft materials, uniaxial tensile testing is performed on dog bone-shaped samples at a strain rate of 0.005 s$^{-1}$ (Instron 5942, Instron, Norwood, MA). This measurement is done both for magnetic soft materials in native conditions and after a heating cycle of 25 at 250 °C (Fig. S21G).

Heat-assisted magnetic programming is utilized for encoding desired magnetization directions in the distributed segments[14]. The magnetic profiles are encoded by locally heating the desired segment around the Curie temperature of the $Nd_2Fe_{14}B$. Particles located at the heated spot lose their magnetization, and their magnetization direction is reoriented by applying an external magnetic field at the desired orientation during the cooling period (Fig. S2C).

Local heating of the magnetized spot area is achieved by utilizing a power-adjustable fiber-couple NIR laser (808 nm, 133 to 457 mW; Edmund Optics, Barrington, NJ) with a collimator (F230SMA-850, Thorlabs, Newton, New Jersey, United States) located at a 7 cm distance. The targeted segment of magnetic soft materials is moved under the laser spot via an automated XY stage (Axidraw v3, Evil Mad Scientist, Sunnyvale, CA) for a heating-cooling cycle. A permanent $Nd_2Fe_{14}B$ magnet (20-mm diameter and 20-mm thickness; Supermagnete, Gottmadingen, Germany) is located at a distance of 20.4 mm to align the magnetization direction of the magnetic particles during the heating-cooling cycle (Fig. S2D, E). Thermal measurements of the heating-cooling cycle are measured by using an infrared thermal camera (ETS320, Wilsonville, OR, USA) at a distance of 7 cm (Fig. S21A). The external magnetic field direction is adjusted relying on the measurements done by a 3D magnetic Hall effect sensor (TLE493D-W2B6, Infineon Technologies, Munich, Germany) located 1 mm below the sample. Desired 3D orientation of the permanent magnet is achieved via stepper motors. The whole magnetization process is automated by a custom script.

Magnetized spot sizes at sample surfaces for heating at varying laser powers and heating times are measured with a magneto-optical sensor (MagViewS, Matesy, Jena, Germany) (Fig. S21B, C). Magnetization and magnetic properties of the magnetic soft materials are characterized by a vibrating-sample magnetometer (VSM; MicroSense, Lowell, MA) (Fig. S21D–F). The Curie temperature measurements are conducted by the comparison of demagnetization ratios after heating hard-magnetized samples to a range of temperatures (Fig. S21D). First, samples are magnetized in VSM under 1.8 T, and their magnetization is measured with VSM. Next, samples are heated to a temperature range of 200 °C–350 °C in an oven. Then, the magnetization of the heated samples is measured via VSM. The demagnetization ratio is calculated by dividing the difference in magnetization strength change after heating to the magnetization strength before heating. Based on these measurements, the Curie temperature is approximated as 260 °C. Magnetic hysteresis loops are obtained by placing a circular sample of 8 mm in diameter on the sample holder and ranging the field between −1.8 and 1.8 T at room temperature. Remanent magnetization ($M_r$) of 83 kA/m and coercivity ($H_c$) of 340 mT are measured (Fig. S21E). Magnetization ratios are determined as the ratio of heat-assisted magnetization strength to hard magnetization strength via VSM under 1.8 T. Heat-assisted magnetic programming ratio is found to be 34.5% for 5 s of heating at 475 mW laser power, resulting in 28.6 kA/m magnetization (Fig. S21F).

The magnetization orientation accuracy of magnetization process is characterized in the range of 0° and 90° in 22.5° increments for in-plane and out-of-plane orientations (Fig. S21H). The orientations of the magnetization calculated from the VSM measurements in x−y axes and y−z axes for in-plane and out-of-plane, respectively. Also, the potential effect of the surrounding magnetized regions on the magnetized spot is found negligible by the comparison of the magnetized spot size for non-magnetized and hard-magnetized samples (Fig. S21I).

## Fabrication and magnetic programming of 2D and 3D structures with multi-material compositions

The 2D and 3D structures with multi-material compositions are fabricated via voxel-assembly method[16]. The voxels of each respective material are fabricated with a 2 mm edge length. First, a positive template of the desired voxel size was fabricated by 3D printing, and

the surface of the template was coated twice with EASE RELEASE™ 200 (Smooth-on, Macungie, PA, USA). Then, Polydimethylsiloxane (PDMS) (a mixture of siloxane base and cross-linking agent at 10:1 mass ratio, Dow Corning, Midland, MI) was poured over the positive template, cured at 60 °C for 6 h in an oven, and peeled off, resulting in a negative template. The surface of the PDMS molds was coated twice with SuperSeal (Smooth-on, Macungie, PA, USA) to prevent sticking and curing inhibition. Subsequently, prepolymers of mixture of Ecoflex 00-30 (Smooth-on, Macungie, PA, USA) and yellow color Silc Pig pigment (Smooth-on, Macungie, PA, USA), Smooth-Sil 960 (Smooth-on, Macungie, PA, USA), mixture of $Nd_2Fe_{14}B$ (MQFP-10-8.5HD-20180, Magnequench, Neomaterials, Toronto, Canada) and Ecoflex 00-30 at a 2:1 ($Nd_2Fe_{14}B$: Ecoflex) mass ratio and mixture of $Nd_2Fe_{14}B$ (MQFP-15-7-20065, Magnequench, Neomaterials, Toronto, Canada) and Dragon-skin 30 (Smooth-on, Macungie, PA, USA) at a 2:1 ($Nd_2Fe_{14}B$: Dragon-skin) mass ratio are cast into molds. The casted materials are degassed under vacuum for 10 min to eliminate any entrapped air, and then the excess materials are scraped away with a flat scraper. Afterward, the prepared samples are cured at room temperature for 4 h. The cured voxels are peeled off by a plastic tweezer with a blunt tip. The prepared magnetic voxels were hard magnetized in VSM under a 1.6 T uniform magnetic field (Fig. S3A).

The fabricated voxels were attached to each other by applying droplets of Ecoflex 00-35 FAST (Smooth-on, Macungie, PA, USA) on the assembly surfaces (Fig. S3B), and cured at room temperature for 5 min. Using the voxel-voxel attachments, the individual layers were formed according to the layer designs provided by our data-driven strategy (Fig. S14C–F). Then, the layers were assembled together sequentially by applying droplets of Ecoflex 00-35 FAST (Smooth-on, Macungie, PA, USA) (Fig. S3C, D) and cured at room temperature for 5 min, resulting in 3D multi-material structures.

### Magnetic actuation and data acquisition setups

Actuation of all the shape-morphing demonstrations (Figs. 1G, 2 and S9) along with the desired morphological task demonstrations (Figs. 3 and S12) is performed using an actuator setup consisting of permanent magnets. Two cylindrical neodymium magnets (60-mm diameter and 10-mm thickness; Supermagnete, Gottmadingen, Germany) are stacked onto each other and fixed on a platform only moving in vertical direction. Controlled vertical movement of the magnet platform is achieved by utilizing a linear motorized stage (LTS300, Thorlabs, Newton, New Jersey, United States). For data acquisition, cameras are located at the side, top, front, and isometric viewpoints. Side, top, and front views are captured by a benchtop digital microscope (Toolcraft USB microscope 5 MP) at 10 fps, while the isometric view is captured by a CMOS camera (Grasshopper 3 USB3, Teledyne Flir, Wilsonville, Oregon, United States) at 60 fps (Fig. S2F).

Magnetic fields for simulation validation and robotic behavior experiments (Figs. 3C, D, 4, S13 and S15) are generated by a custom 3-axis Helmholtz coil setup that can generate a maximum uniform magnetic field of 13 mT in all axes within a $4 \times 4 \times 4\ cm^3$ workspace (Fig. S2G). Data acquisition at 175 fps is realized by two cameras (Basler aCa2040-90uc, Ahrensburg, Germany) placed at the side and isometric views.

### Performance measurements

The performances of the designs are measured through the pre-defined performance objective functions. While the average positional root means square error (RMSE) is utilized for all the shape-matching demonstrations, the robotic behavior demonstrations have different performance objective functions (SI S5.1). For the shape-matching demonstrations, the average positional RMSE is calculated considering the voxel position differences between the desired shape and the simulation results. The desired shape definitions are provided in SI S5.2.

### Design space calculation

The design spaces for different demonstrations are quantified to provide a better understanding of the relative and increasing complexity. The reported design spaces in Table S3 are calculated by discretizing the continuous magnetization direction of segments. Then, a simple calculation considering all the possibilities of the voxel and magnetization directions yields the rough estimation of the design space (SI S6). Figures S14C–F and S22A–F show the close-up morphology and magnetic profile design of demonstrations. The parameter details of the voxel, segment, and parameter numbers and the calculated design space resulting from the pre-defined workspace for these demonstrations are gathered in Tables S3 and S5.

## Data availability

All data generated or analyzed during this study are included in the published article and its Supplementary Information. The algorithm run data[54] generated in this study have been deposited in the Zenodo database under accession code https://doi.org/10.5281/zenodo.14827843.

## Code availability

The source code of the developed simulation and design algorithm[55] has been deposited in the Zenodo under accession code https://doi.org/10.5281/zenodo.14827691. It is also available at the link: https://github.com/AlpKaracakol/data_driven_magnetic_soft_material_design.

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

## Acknowledgements

The authors thank Dr. Utku Culha for initial discussions, Cem Kaya for helping with the experimental setup, Anitha Shiva for helping with magnetic characterizations, Muhammad Yunusa for helping with the tensile testing measurements, and Dr. Ugur Bozuyuk, Dr. Gaurav Gardi, and Dr. Mehmet Efe Teriyaki for the discussions. This work was funded by the Max Planck Society and European Research Council (ERC) Advanced Grant SoMMoR project with grant no: 834531. A.C.K. thanks the ETH and Max Planck Center for Learning Systems for the funding during a part of this work.

## Author contributions

A.C.K. participated in the study design, simulation and algorithm development, experimental procedures, data collection, data analysis, and manuscript writing. Y.A. participated in the study design, data analysis, research supervision, and manuscript writing. S.O.D. assisted with the experimental procedures and data collection. M.S. participated in the study design, research supervision, and manuscript writing.

## Funding

## Competing interests

The authors declare no competing interests.
