## [Transparent Peer Review file · Nature Communications]

Data-driven design of shape-programmable magnetic soft materials

Corresponding Author: Professor Metin Sitti

Version 0:

Reviewer comments:

Reviewer #1

(Remarks to the Author)

The manuscript entitled “Data-driven design of shape-programmable magnetic soft materials” reports a data-driven design strategy to search the design space of distributed spatial magnetization profile and morphology for desired shape-morphing and behavior of magnetic soft materials. Compared to the conventional data-based method, the proposed design strategy expands the design space of the magnetization orientation from 2D to 3D, thus enabling more flexibility in the deformation prediction of magnetic soft materials. However, the programming of 3D magnetization directions presents greater challenges for material preparation. Some detailed comments are as follows.

1. As mentioned in the manuscript, the curie temperature of the NdFeB particle is about 260°C. Will this temperature influence the mechanics properties of the Ecoflex matrix?
2. The modulus and magnetic remanence of magnetic soft materials are fitted to match the experimental and simulation results. Does this mean that the proposed method cannot effectively predict the deformation of real materials?
3. Does the desired magnetization orientation affect the remanence of the magnetic materials? Can the final magnetization orientation actually achieve the desired value? It would be helpful to include experimental results to validate the effectiveness and stability of the reprogramming method. Relying on experimental results to determine material parameters repeatedly will render the proposed method pointless.
4. Why can't the intuitive design in Fig. 3B achieve jumping? Does the design in Fig. 3D sacrifice other performance aspects, like deformation capability, in order to enable jumping?
5. A magnet is used to realign the magnetic particles. Why choose 20.4mm distance? Which properties of the magnetic materials are affected by this distance?
6. When does the magnetic field transition from positive to negative in Fig. 3B? Other figures also lack important information, making it hard to grasp their content without closely reading the text. The authors are suggested to reorganize all figures for clearer presentation to a general audience. For example, in Fig.1F, the filled voxel ratio and net magnetization can be reflected in the figure.
7. What if design 3D structures without soft materials? The jumping and traversing locomotion can also be achieved using pure rigid materials.

(Remarks on code availability)

Reviewer #2

(Remarks to the Author)

The manuscripts named “Data-driven design of shape-programmable magnetic soft materials” explores the design space of distributed magnetization profile and desired morphology relying on stochastic design alterations guided by a predictive neural network. The data-driven design method is demonstrated to enhance shape morphing capabilities, optimize their morphology and improve jumping performance of the magnetic soft robots. Please find below the comments from the reviewer to help improve the manuscript.

- 1)The first part of the introduction does not seem to align well with the main research focus of the paper. The proposed data-driven method primarily targets magnetic soft materials, so it is unnecessary to delve too deeply into the general aspects of

soft materials.

2)The introduction discusses the spatial programming capabilities of magnetic materials, but it contains a lot of qualitative analysis without the necessary references.

3)As shown in Figure 1B, the resolution of morphological features and distributed magnetization are represented by voxel and segment. How do we define the number and size of voxels and segments? Do they influence the optimization results ?

4)In the design generation and performance prediction, how did the authors define the required threshold and maximum generation number? Were these based on empirical judgment or experimental determination?

5)The reviewer does not fully agree that the deformation shown in Figure 2 cannot be achieved through intuitive design. As many theoretical studies, such as, doi: 10.1109/TASE.2023.3313395; doi: 10.1016/j.ijengsci.2020.103391; doi: 10.1016/j.jmps.2021.104739 have already developed deformation models for hard magnetic soft materials. These models can accurately predict deformation outcomes under various magnetization designs. Therefore, what advantages does the current data-driven approach offer in shape prediction compared to these existing models?

6)Is it possible for the current method to achieve the desired morphology solely through the optimization of the magnetization profile? In some cases, while the optimized designs of magnetically-driven soft robots lead to more complex deformation, they also increase manufacturing difficulty and may reduce output force.

7)Based on the results presented in the paper, the application of the current data-driven design method relies heavily on specific manufacturing techniques. Beyond heat-assisted programming and manual assembly, could this approach be extended to other manufacturing methods? For example, techniques such as template-assisted methods, photolithography, or 3D printing?

8)Can the manufacturing resolution of heat-assisted programming and manual assembly methods meet the demands of the optimized design? Both methods are prone to unavoidable manufacturing errors, could this impact the experimental results? Additionally, will the magnetic properties of the magnetic soft materials experience any degradation after heat programming?

9)The reviewer is unclear about how the author defines and calculates the design space of the current method.

10)As the author mentioned, existing designs can utilize the various morphological changes of magnetic soft materials or manipulate the distribution of external magnetic fields to achieve many complex locomotion. In comparison, what is the necessity and advancement of using data-driven design methods? Could the author provide a comparison of the improvements in functionality and movement complexity of magnetically-driven robots under the current design methods?

11)The jumping experiment shown in Figure 3 is interesting. Is it possible to control the jumping direction through data-driven design?

12)Design of multi-material and 3D magnetic soft millirobots is a creative idea. Zhang, J. et al. (doi.org/doi:10.1126/scirobotics.abf0112) assembled magnetic voxel units heterogeneous, demonstrating very complex locomotion such as rolling, rotation, and bistability, which can facilitate functions like drug transport and release. The current approach uses data-driven methods to assemble magnetic voxels, however, the results only show the jumping movements, which do not effectively illustrate the advantages of the current method. Could the author include additional applications to better demonstrate the benefits of the data-driven design?

(Remarks on code availability)

I have read and successfully run the author's code on my computer.

Reviewer #3

(Remarks to the Author)

The authors developed a data-driven design strategy for magnetic soft materials to achieve diverse 2D and 3D shape morphing and locomotion. The design framework is established by spatially programming the magnetization profile, morphology, and multi-material composition of magnetic soft materials, allowing them to achieve desired static or dynamic shape morphing behaviors under pre-defined external magnetic fields. Various demonstrations, including 2D/3D shape morphing of magnetic soft beams, jumping behaviors of magnetic soft millirobots, and controlled locomotion of multi-material 3D magnetic soft millirobots, showcase the robustness of the design strategy. The work is interesting, and the proposed design framework appears to be highly effective. I recommend this work for publication in Nature Communications after the following issues are addressed:

1. The authors used a mass-spring lattice model coupled with magnetic forces and torques as a simulation tool to support the data-driven design. However, no details on the implementation of the model are provided through reference provided. Supplementary examples should be provided regarding the static and dynamic models.
2. With an analytical model available, can the author discuss more about why stochastic optimization is chosen instead of gradient-based training of a neural network for the problems in the paper?
3. The optimization is less effective when predicting those with sharp turns in Fig. 2C and Fig. 2E. Is it possible to evaluate and decide the minimum radius of curvature you can have?

4. The desired static and dynamic shape morphing of magnetic soft materials is achieved by optimizing their magnetization and morphology. For shape morphing magnetic soft beams, the optimized designs often contain holes and imperfections. Is it possible to achieve these complex shape morphing using a smooth, perfect beam by combining the data-driven algorithm with theoretical beam models?

5. A marginal change in the beam structure and magnetization can lead to dramatic configuration differences. Is this related to the snap-through of slender magnetic beam structures? For a given beam structure like the one in Fig. S1A, would applying the magnetic field at different speeds or different paths with the slight out-of-plane component cause the beam to deform into a different configuration in Fig. S1B?

6. The author showed the marginal effect that a small change in beam length can cause drastic deformation differences. For the 3D dynamic robot, would the manual assembly of each voxel lead to fabrication errors and behaviors different from targeted ones?

7. The author's group has already demonstrated various soft robotics with incredible moving capabilities by rolling or tumbling beam structures. Fig.4 demonstrates multiple 3D robot designs with much more complicated structures but weaker moving performance. How would the authors justify the merits of using the current design framework to design these 3D and multilateral robots?

8. There is a typo in the manuscript. In the third paragraph of the Section "Jumping behavior for magnetic soft millirobots", the figure citation "Fig. 10B" should be "Fig. S10B". The readability of SI can be improved.

(Remarks on code availability)

Version 1:

Reviewer comments:

Reviewer #1

(Remarks to the Author)

The authors have addressed my comments.

(Remarks on code availability)

Reviewer #2

(Remarks to the Author)

All my concerns have been addressed, I recommend this paper was published in the journal.

(Remarks on code availability)

Reviewer #3

(Remarks to the Author)

I have reviewed the revised draft of the manuscript titled Data-driven design of shape-programmable magnetic soft materials, and I am pleased to confirm that all my questions and concerns have been adequately addressed. I recommend its publication.

(Remarks on code availability)

Response Letter

RE: *Nature Communications*: Revised version of manuscript NCOMMS-24-61701 “Data-driven design of shape-programmable magnetic soft materials”

We sincerely thank the reviewers for their detailed feedback and constructive suggestions which have significantly improved the clarity, and quality of our manuscript and enhanced our ability to communicate the novelty of our work. For the convenience of the reviewers, we have indicated the added and updated text in the manuscript and SI with **blue font color**.

In the following, we have addressed the reviewer comments point by point **in bold**, where additions and updates in the manuscript are shown in **blue**.

Point-by-point reviewer comment responses:

Reviewer #1 (Remarks to the Author):

The manuscript entitled “Data-driven design of shape-programmable magnetic soft materials” reports a data-driven design strategy to search the design space of distributed spatial magnetization profile and morphology for desired shape-morphing and behavior of magnetic soft materials. Compared to the conventional data-based method, the proposed design strategy expands the design space of the magnetization orientation from 2D to 3D, thus enabling more flexibility in the deformation prediction of magnetic soft materials. However, the programming of 3D magnetization directions presents greater challenges for material preparation. Some detailed comments are as follows.

Response: We thank the reviewer for the constructive feedback. We have carefully reviewed the reviewer’s comments and addressed all the concerns and queries.

1. As mentioned in the manuscript, the curie temperature of the NdFeB particle is about 260°C. Will this temperature influence the mechanics properties of the Ecoflex matrix?

Response: We thank the reviewer for this comment. Ecoflex 00-30 has been shown to be thermally stable up to over 300°C¹. In our work, the heated regions are exposed to 250°C for only 5 seconds (Fig. S17A) and our characterization of the material after 25 heating-cooling cycles showed negligible changes in Young’s modulus (Fig. S17G). Since each magnetized region in this study is heated only once, we assumed that any changes in the mechanical properties are negligible.

Curie temperature of the NdFeB powder is specified as 219°C in its technical datasheet². However, our thermomagnetic characterization revealed that the specific batch used in this study had a Curie temperature ~260°C (Fig. S17D). We acknowledge that a subsequent batch we received aligns with the advertised Curie temperature of 219°C. For future publications, we plan to utilize this new batch.

2. The modulus and magnetic remanence of magnetic soft materials are fitted to match the experimental and simulation results. Does this mean that the proposed method cannot effectively predict the deformation of real materials?

Response: We thank the reviewer for this comment. The effectiveness of the simulation engine adapted in this work is supported by numerous studies that have demonstrated its applicability to a wide range of materials³⁻⁶. Furthermore, our deformation predictions, even without parameter fitting, show good agreement with experimental results, particularly for applied magnetic fields above 3 mT, as illustrated in Fig. S16A and Fig. S16C. This highlights that the computational predictive model is inherently robust and can be applied effectively without parameter fitting to predict deformations.

The primary limitation of the unfitted model is the overprediction of deformations in the low magnetic field range below 3 mT. While magnetic actuation fields employed in our demonstrations were equal or above 10 mT, parameter fitting is primarily employed to enhance accuracy in starting phase of actuation, ensuring better simulation to experiment transfer.

3. Does the desired magnetization orientation affect the remanence of the magnetic materials? Can the final magnetization orientation actually achieve the desired value? It would be helpful to include experimental results to validate the effectiveness and stability of the reprogramming method. Relying on experimental results to determine material parameters repeatedly will render the proposed method pointless.

Response: We thank the reviewer for these comments. Magnetic remanence of our material is invariant to the desired magnetization orientation. Since the magnetic particles are randomly dispersed within the polymer matrix, there is no domain alignment prior to magnetization, preventing any directional anisotropy. As shown in Fig. S17H, programmed magnetization orientation closely matches with the desired orientation. Following the reviewer's suggestion, we have further experimentally measured the magnetic remanence at different magnetic orientations, showing statistically negligible differences (Fig. R1):

Figure R1. Magnetization strength at different desired orientations. The magnetization strength comparison for desired magnetization orientation of heat-assisted magnetization method characterized in the range of 0° to 90° in 22.5° increments. The points represent individual sample results, the center horizontal line depicts the mean and the error bars show the one-sigma range of standard deviation. No statistically significant difference in magnetization strength was observed among groups ($p > 0.05$, one-way ANOVA with Tukey's multiple comparisons test)

We believe that characterizing the properties of materials fabricated and programmed via non-commercial/conventional methods and without off-the-shelf products is essential for reproducibility in this field. Even variations between different batches of commercial products can significantly impact experimental results, as highlighted above for the Curie temperature of NdFeB batches. This is why we conducted extensive characterization and validation experiments to ensure robustness and reproducibility, which we hope will be a standard practice in cutting-edge research.

4. Why can't the intuitive design in Fig. 3B achieve jumping? Does the design in Fig. 3D sacrifice other performance aspects, like deformation capability, in order to enable jumping?

Response: We thank the reviewer for this query. To objectively compare the performance of our data-driven strategy, we adapted an intuitive design and external magnetic field signal employed for jumping from the literature⁷. Only difference was the 3-times smaller magnetization strength of our material compared to the magnetic particles used in the previous report⁷. This reduced magnetization strength is why the intuitive design in Fig. 3C cannot achieve jumping in our work under the same experimental conditions applied in the previous study.

As with any optimization process, performance maximization with a single objective (e.g., jumping height) involves trade-offs among various parameters. During this process, certain performance aspects—potentially unobservable to human intuition—may be slightly reduced to optimize the primary objective. However, in this demonstration, the deformation capability was comparable in both designs, as evidenced by the average strain energy density calculated in our simulation engine (Fig. 3D).

5. A magnet is used to realign the magnetic particles. Why choose 20.4mm distance? Which properties of the magnetic materials are affected by this distance?

Response: We thank the reviewer for this comment. Among the two fabrication methods utilized in this work, the heat-assisted magnetic programming method employs a magnet to align the magnetization direction of the magnetic particles, which are physically fixed within the polymer matrix. The 20.4 mm distance ensures magnetic field values at the desired magnetization spot range between approximately 26 mT and 45 mT. Given that the coercive field of the NdFeB particles used in this study is around 340 mT, these field values remain significantly below the coercive field, preventing undesired changes to the samples' magnetization.

6. When does the magnetic field transition from positive to negative in Fig. 3B? Other figures also lack important information, making it hard to grasp their content without closely reading the text. The authors are suggested to reorganize all figures for clearer presentation to a general audience. For example, in Fig. 1F, the filled voxel ratio and net magnetization can be reflected in the figure.

Response: We thank the reviewer for the valuable feedback. In Fig. 3B, the magnetic field transitions from positive (upward z direction) to negative at the time stamp indicated as 0 ms. This has been clarified in the figure as recommended by the reviewer. Similarly, in Fig. 1F, we have incorporated visual indicators for the filled voxel ratio and net magnetization to better reflect this information directly in the figure, as suggested by the reviewer.

Additionally, we have reviewed all figures in the manuscript to improve their clarity and accessibility. Necessary additions and clarifications have been made as listed below:

- In Fig. 1F, the Feature 1 and Feature 2 have been updated to “Filled voxel ratio” and “Net magnetization direction,” respectively.
- In Fig. 1G, further captions of “morphology”, “magnetization direction with arrow”, “simulation” and “experiment” have been added.

- In Fig. 3B-D, Fig. 4C, D, Fig. S10A, B, and Fig. S12, the magnetic field change instant has been clarified with labels at $t=0^-$, $t=0$ and $t=0^+$ instants, following conventions from signal processing and control systems.
- In Fig. S1A, a “Simulation” caption has been added to indicate that the result is from the simulation environment.
- In Fig. S2A, captions for “magnetic soft elastomer” and “200 μm tapes” have been added to clarify the molds.
- In Fig. S2F, cartesian reference axis is added, for further clarity.
- In Fig. S3D, captions have been added to label the fabricated robot and the numbers on the right side of the robot structures.
- In Fig. S4B, a “pseudo-code” title has been added for clarity on the given algorithm.
- In Fig. S6A, B, captions for “Desired shape” and “Simulation” have been added for improved clarity.
- In Fig. S7A, the missing legends for morphology and magnetization arrow notations have been added.
- In Fig. S8A-L, additional figure descriptions, and references for the demonstrations from this work or state-of-the-art works in literature have been included to facilitate comparisons between the shapes in Figures S8C-L and S8B.
- In Fig. S9, missing legends for morphology and magnetization arrow notations have been added.

7. What if design 3D structures without soft materials? The jumping and traversing locomotion can also be achieved using pure rigid materials.

Response: We thank the reviewer for this intriguing query. We agree with the reviewer that locomotion modes like jumping and traversing can indeed be achieved using purely rigid materials. The proposed data-driven design method could, in principle, be adapted for designing purely rigid structures as well. However, our method is specifically developed to address the unique challenges associated with designing soft materials and hybrid systems combining soft and rigid components, as highlighted in Fig. 4, Fig. S11 and Fig. S14. These challenges include handling the nonlinearity, high deformability, and complex interactions intrinsic to soft materials, and our focus in this work is on overcoming these complexities to advance soft material design.

Reviewer #2 (Remarks to the Author):

The manuscripts named “Data-driven design of shape-programmable magnetic soft materials” explores the design space of distributed magnetization profile and desired morphology relying on stochastic design alterations guided by a predictive neural network. The data-driven design method is demonstrated to enhance shape morphing capabilities, optimize their morphology and improve jumping performance of the magnetic soft robots. Please find below the comments from the reviewer to help improve the manuscript.

Response: We thank the reviewer for the valuable comments and suggestions. We have addressed the reviewer’s comments point-by-point as detailed below.

1) The first part of the introduction does not seem to align well with the main research focus of the paper. The proposed data-driven method primarily targets magnetic soft materials, so it is unnecessary to delve too deeply into the general aspects of soft materials.

Response: We thank the reviewer for this feedback. While the general aspects of soft materials were included in our original submission to provide broader context, we understand that delving too deeply into them may detract from the focus on magnetic soft materials and the proposed data-driven method. To address this, we have revised the introduction based on the reviewer’s suggestion, as following:

“Stimuli-responsive materials, which respond to external stimuli such as light, pH, and magnetic or electrical fields⁸⁻¹¹, have gained significant attention for enabling complex shape-morphing in untethered structures¹²⁻¹⁴, and shown distinct advantages in fields like object manipulation, soft robotics, wearable devices, and biomedical applications^{8,9,15-17}. Among various responsive composites, magnetic soft materials stand out due to their rapid advancements in micron-scale resolution, three-dimensional (3D) directionality, multi-material compositions, and complex 3D structures¹⁸⁻²³. These capabilities enable a wide range of static and dynamic shape-morphing behaviors across different length scales, from micro to milli.

Magnetic soft materials consist of hard magnetic particles (e.g., $\text{Nd}_2\text{Fe}_{14}\text{B}$, CrO_2) embedded within a soft material matrix (e.g., silicone rubber, polydimethylsiloxane (PDMS), hydrogels) and spatially magnetized at desired orientations to create a distributed magnetic moment. When subjected to an external magnetic field, the particles experience torques that align their magnetization with the field, generating programmable shape deformations. This spatial programmability is achieved through techniques such as jig-assisted assembly¹⁸, lithography^{19,22}, 3D printing²⁰, and local heat-assisted magnetization in micromachined structures²¹. These methods allow precise control over material composition, magnetization orientation and magnitude, and structural features, enabling complex shape-morphing capabilities.”

2) The introduction discusses the spatial programming capabilities of magnetic materials, but it contains a lot of qualitative analysis without the necessary references.

Response: We thank the reviewer for highlighting this important point. To address this, we have incorporated relevant references^{11,24,25} regarding the coupling effects between magnetic and mechanical responses and the challenges of non-intuitive shape-morphing in magnetic soft materials.

3) As shown in Figure 1B, the resolution of morphological features and distributed magnetization are represented by voxel and segment. How do we define the number and size of voxels and segments? Do they influence the optimization results?

Response: We thank the reviewer for this comment. The resolution of voxels and segments influences the optimization process by affecting the dimensionality and granularity of the design space. While finer resolutions allow for more detailed and expressive designs, they also increase computational complexity and may result in solutions that are more challenging to fabricate. Conversely, coarser resolutions simplify the optimization process and improve computational efficiency but may limit the diversity or precision of the resulting designs. In this work, these parameters are defined depending on the capabilities and/or limitations of the fabrication method along with the consideration of the computational time of the simulations as followed:

- The voxel sizes are defined as 200 μm and 2 mm for the demonstrations fabricated by heat-assisted magnetic programming and voxel-assembly methods, respectively.
 - Heat-assisted magnetic programming: A voxel size with a side length of 200 μm was used due to the thickness of our magnetic soft material, which was chosen to ensure uniform magnetization throughout the volume via heat-assisted magnetization (Fig. S17C).
 - Voxel-assembly: A voxel size with a side length of 2 mm was chosen to minimize fabrication errors and complexity, as manual gluing of smaller voxels increases artifacts and fabrication time.
- The segment size, thus voxel number within a segment, is also chosen depending on the capabilities of the fabrication methods.
 - Heat-assisted magnetic programming: The magnetization resolution of the method in our setup is at 600 μm due to the laser spot size. Considering this constraint, the segment sizes are selected at smallest 600 μm in length and width. For the magnetic soft beam demonstrations, segments with 600 μm in length and 1 mm in width, which are equivalent to the width of the beam, are employed. For the magnetic soft sheets, segments with 600 μm in length and width are used, only except for Fig. S9B, in which segments of 1 mm length and width were chosen to meet integer division requirement.
 - Voxel-assembly: Since each voxel is magnetized one by one in voxel-assembly method, the segments are equivalent to a single voxel for all the demonstrations with 2 mm voxel size.
- Total voxel numbers of the demonstrations are defined by considering either the computational evaluation time or the expected average fabrication time of the samples.
 - For 200 μm voxels, the computational evaluation time was targeted to fall within the 60–120 seconds range.
 - For 2 mm voxels, the voxel number was limited to approximately 250, based on the fabrication constraints of the voxel-assembly method. This limit assumes a fabrication time of ~1 minute per voxel for manual gluing, allowing the final design to be completed within ~6 hours.

Based on the reviewer's comments, we have now included the discussion above to our revised supplementary information under SI S1.1 with the subtitle "Choosing voxel and segment resolutions for data-driven design algorithm".

4) In the design generation and performance prediction, how did the authors define the required threshold and maximum generation number? Were these based on empirical judgment or experimental determination?

Response: We thank the reviewer for this query. In our framework, we have set the maximum number of generations as the only threshold. The maximum number of 5000 generations for most of the demonstrations was chosen based on computational feasibility and the simulation evaluation budget. For a simulation time of approximately 60–120 seconds per evaluation, we targeted a total runtime of one week for practical applications. This generation limit is halved to 2500 for the sheet demonstrations in Fig. S9B and Fig. S10B due to the severe computational times. In order to show that the framework continuously improves on the designs without trapping into local optima, we ran the algorithm longer for three specific cases representing different structural and material complexities (Figs. S2B, S7C, and S4E). The only exception for which we ran a lower number of generation (2000 generations) is for the demonstration in Fig. S7A, where the run was ended due to a technical problem in the cluster environment.

Based on the reviewer's comments, we have included performance evolution of the 2D and 3D shape-morphing magnetic soft beams (Fig. S10, see below), as well as, additional information regarding the definition of the threshold into the supplementary information under the SI S1.2 at the end of the subtitle "Implementation of the overall algorithm".

In the revised manuscript:

"Nevertheless, performance evolution graphs show initial rapid improvements, followed by a slower but steady increase in performance for both 2D and 3D shape-morphing cases, which could alleviate these deviations with extended algorithm runs (Fig. S10)."

In the revised supplementary notes:

"In our framework, we have set the maximum number of generations as the only threshold. The maximum number of 5000 generations for most of the demonstrations was chosen based on computational feasibility and the simulation evaluation budget. For a simulation time of approximately 60–120 seconds per evaluation, we targeted a total runtime of one week for practical applications. This generation limit is halved to 2500 for the sheet demonstrations in Fig. S12B and Fig. S13B due to the severe computational times. In order to show that the framework continuously improves on the designs without trapping into local optima (Fig. S10), we ran the algorithm longer for three specific cases representing different structural and material complexities (Figs. S2B, S9C, and S4E)."

Figure S10. Evolution of performance for 2D and 3D shape-morphing magnetic soft beams during the data-driven design progress. (A, B) Evolution of the shape-morphing behavior for 2D (A) and 3D (B) magnetic soft beams with the performance in the range of [0, 1] and the evaluation range of [0, 2.5e5]. Both cases show initial rapid improvements, followed by a slower but steady increase in performance. (C) Extended evolution of 2D shape-morphing (square) over 5e5 evaluations and 3D shape-morphing (helix) over 7.5e5 evaluations, further illustrating the steady performance gains.

5) The reviewer does not fully agree that the deformation shown in Figure 2 cannot be achieved through intuitive design. As many theoretical studies, such as, doi: 10.1109/TASE.2023.3313395; doi: 10.1016/j.ijengsci.2020.103391; doi: 10.1016/j.jmps.2021.104739 have already developed deformation models for hard magnetic soft materials. These models can accurately predict deformation outcomes under various magnetization designs. Therefore, what advantages does the current data-driven approach offer in shape prediction compared to these existing models?

Response: We thank the reviewer for this thoughtful question and for highlighting relevant theoretical studies. While these aforementioned models²⁶⁻²⁸, as well as, other theoretical models^{25,29,30} are capable of predicting deformation under predefined magnetization profiles, they are limited in scope to simplified 1D/2D magnetization profiles with simple rod or beam shapes. However, as shown in our manuscript, increased complexity in distributed magnetic profile and morphology can result in intricate shape-morphing configurations. Furthermore, minute local changes within the magnetic profile can dramatically affect the resulting deformations, as shown in our original manuscript (Fig. S6). To further investigate the sensitivity of shape-morphing based on changes of magnetization direction within specific segments, we generated a heat-map of shape-morphing change ratio by varying the magnetization direction in increments of 15° from -180° to 180°, in all 20 different segments (Fig. S7, see below). The resulting heat-map shows the inherently highly non-linear design space with numerous local minima both with different segment numbers and magnetization directions.

Furthermore, our data-driven design approach addresses a much broader range of complex design challenges and offers distinct advantages that address limitations of these models:

1. Design Exploration Beyond Intuition:

Theoretical models assume a known magnetization profile and solve forward problems to predict deformation. However, identifying an appropriate magnetization pattern for a

desired target deformation or behavior often requires significant experience and manual effort, particularly for high-dimensional problems. Our framework automates this process, exploring a vast design space to uncover solutions that may be infeasible or difficult to derive using intuition alone.

2. Coupled Optimization of Morphology and Magnetization:

Existing models typically rely on only the magnetization profiles, which limits their ability to address scenarios requiring simultaneous optimization of structural morphology and distributed magnetization. Our data-driven framework, in contrast, explores both morphology and magnetization concurrently, enabling the discovery of designs that achieve complex, non-intuitive deformations and behaviours.

3. Broader Applicability:

Our method extends far beyond the 2D shape-morphing of magnetic beams. It is applicable to a wide variety of scenarios, including 3D shape-morphing of beams, transformation of sheets, and tasks ranging from morphological optimization to robotic functionalities. This includes the design of 2D and 3D robots with multi-material compositions, as well as configurable, multi-responsive robots. These capabilities are demonstrated through the diverse examples showcased across our figures.

In summary, while existing theoretical models are effective for predicting deformation in simplified scenarios, our data-driven approach offers broader applicability, automated exploration of non-intuitive designs, and coupled optimization of morphology and magnetization, addressing the inherent complexities of real-world design challenges.

Based on the reviewer's comment, we have included the following statements and the figure in the revised manuscript and supplementary notes, respectively:

“We further validated the non-intuitive nature of the coupled design space by introducing marginal changes in magnetization and morphology of the sinusoidal shape-morphing beam, as well as the external magnetic field, which resulted in dramatic shifts from the desired shape in the simulation environment (Figs. S6-S8). These sensitivity analysis results show that the design space is inherently non-linear with numerous local minima and there are a wide range of equilibrium states for a given design, which depends on the initial design state (Figs. S7) and the induced magnetic field input sequences (Fig. S8) to the system.”

Figure S7. Sensitivity heatmap and selected examples for the effect of magnetization on shape-morphing of magnetic soft material. (A) The sensitivity heatmap showing the shape-morphing change ratio (%), for the demonstration in Fig. 2A, based on the relative change of the magnetization direction (θ , y-axis) at different segment numbers (x-axis). The heatmap and illustrative cases highlight the non-linear and non-intuitive behavior of magnetic

soft materials. The selected examples are marked with orange boxes, and their corresponding shape-morphing results are linked via orange lines. Shape-morphing change ratio is calculated as the average displacement error, with respect to the original design, normalized to the body length. Color bars represent the shape-morphing change ratio (%) and average strain energy density (J/m^3). **(B)** The shape-morphing of the original design is shown for comparison, along with the description of the relative rotation change (θ), and the respective segment numbers of the given original design.

6) Is it possible for the current method to achieve the desired morphology solely through the optimization of the magnetization profile? In some cases, while the optimized designs of magnetically-driven soft robots lead to more complex deformation, they also increase manufacturing difficulty and may reduce output force.

Response: We thank the reviewer for raising this question. The optimization of magnetization profiles alone can achieve desired morphologies in certain cases, particularly for relatively simple target shapes or tasks. However, as the complexity of the desired shape-morphing or behavior increases, the coupling between morphology and magnetization becomes more significant, and the simultaneous optimization of both is typically necessary to fully exploit the design space and achieve high-performance results.

It is worth noting that the flexibility of our framework allows users to define design objectives and constraints, such as prioritizing simpler designs to ease manufacturing or maintaining sufficient output force for functional performance. This ensures the generation of designs that balance these trade-offs effectively, as demonstrated in our diverse examples. Ultimately, while magnetization-only optimization can be applied for specific straightforward tasks, it is the combined approach enables the complex shape-morphing and robotic functionalities.

7) Based on the results presented in the paper, the application of the current data-driven design method relies heavily on specific manufacturing techniques. Beyond heat-assisted programming and manual assembly, could this approach be extended to other manufacturing methods? For example, techniques such as template-assisted methods, photolithography, or 3D printing?

Response: We thank the reviewer for this insightful query. Our demonstrations have already highlighted the adaptability of the framework across different scales and fabrication techniques, including voxel-based manual assembly and heat-assisted magnetic programming. This versatility indicates that extending to other methods, would primarily involve practical implementation adjustments rather than fundamental changes to the framework itself.

For example, in our framework, the methods mentioned by the reviewer could be implemented as:

- Magnetization patterns compatible with template-assisted manufacturing¹⁸ could be mapped to the voxel-based representation, and the designed complex structural designs can be molded or laser cut.
- The resulting 2D structural design could be translated to photolithography^{19,22} masks, and the potential magnetization restrictions could be involved in the representation part.

- **The layer thickness inherent to 3D printing methods²⁰ can be defined as voxel size in our framework, ensuring that the optimized designs align with the layer-by-layer resolution of the 3D printer, and the potential magnetization restrictions could be involved in the representation part.**

8) Can the manufacturing resolution of heat-assisted programming and manual assembly methods meet the demands of the optimized design? Both methods are prone to unavoidable manufacturing errors, could this impact the experimental results? Additionally, will the magnetic properties of the magnetic soft materials experience any degradation after heat programming?

Response: We thank the reviewer for this comment. We agree with the reviewer that the manufacturing artifacts have an impact on the transfer of the optimized designs from simulation to experiments (sim2real). Being said that, the heat-assisted magnetic programming method is characterized in detail in terms of both the fabrication of the desired structure and the magnetic programming in desired magnetization directions as shown in Fig. S17. Furthermore, this method minimizes the handling errors, since it relies on laser-cutting for the structural fabrication, and automated magnetic programming via a precise stage, and well-characterized laser-heating (Fig. S2A-E). The quantified comparison between the simulation and experimental results for 2D shape-morphing demonstrations, presented in Table S2, reveals an average difference of approximately 3% of the body length (~340 μm).

For the voxel-assembly method we acknowledge that the fabrication involves manual handling of the samples at every step of the fabrication (FIG. S3), making it highly prone to fabrication artifacts. To minimize potential handling and fabrication errors, we have employed voxels with a side length of 2 mm, resulting in negligible imperfections on the assembled structure. Overall, the successful utilization of both manufacturing methods validates that our framework can be extended to other fabrication methods, and varying size scales.

The magnetic properties of the magnetic soft materials do not experience degradation on its core magnetic properties. This stability is due to the well-studied Curie temperature phenomenon, which is extensively understood and widely utilized in industrial applications over the last century, particularly in magneto-optical data storage technologies³¹⁻³³. Our approach leverages this phenomenon to erase magnetization at the heated spot by raising its temperature near the Curie point. During the cooling phase, a controlled, relatively low magnetic field is applied to re-align the magnetization in the desired direction, ensuring precise and reliable programming of the material²¹.

9) The reviewer is unclear about how the author defines and calculates the design space of the current method.

Response: We thank the reviewer for this comment. The design space in our framework is defined as the set of all possible configurations of structural design and distributed magnetization directions, which are optimized to achieve desired deformation or functionality. Structural design is defined by the presence or absence of voxels in a grid, determining the physical structure's shape and arrangement. Distributed

magnetization is represented by the magnetization direction assigned to each voxel or segment. Following the description provided in SI S6;

The total design space size (Ω) can be roughly approximated by:

$$\Omega = (V)^n (S)^m,$$

Where:

- **V:** The number of material types, including the "void" option for empty voxels,
- **n:** The total number of voxels defining the structural design,
- **S:** The number of discretized magnetization directions available for each segment,
- **m:** The number of segments for magnetization programming.

Following these definitions, these parameters could be calculated via the help of the provided parameters in Table S5 as,

- **V = number of materials + 1**
- **n = Voxel number in X axis * Voxel number in Y axis * Voxel number in Z axis**
- **S depends on the magnetic programming method**
 - In heat-assisted magnetic programming,
 - For 2D demonstrations with 1° discretization, **S = 360**,
 - For 3D demonstrations with 1° discretization, **S = 360*180 = 64800**.
 - In voxel-assembly method **S** is restricted to 6 discrete directions in Cartesian axes, thus **S = 6**.
- **m = Segment number in X axis * Segment number in Y axis * Segment number in Z axis**

As an example, for the demonstration in Fig. 4, from the Table S5, we can obtain,

- **V = 4 + 1 = 5**
- **n = 7 * 7 * 5 = 245**
- **S = 6 (voxel-assembly fabrication)**
- **m = 7 * 7 * 5 = 245**

Thus, the design space size,

- **$\Omega = (5)^{245} (6)^{245} \approx 7.8e361$**

The resulting design space sizes for all the demonstrations can be found in Table S3 that are calculated as explained in the example via the parameters obtained from Table S5. Based on the reviewer's suggestion, we have now expanded our explanation of design space calculation at the revised supplementary information (SI S6).

10) As the author mentioned, existing designs can utilize the various morphological changes of magnetic soft materials or manipulate the distribution of external magnetic fields to achieve many complex locomotion. In comparison, what is the necessity and advancement of using data-driven design methods? Could the author provide a comparison of the improvements in functionality and movement complexity of magnetically-driven robots under the current design methods?

Response: We thank the reviewer for this thought-provoking question. To objectively compare the performance of our data-driven strategy, we adapted an intuitive design and external magnetic field signal employed for jumping from the literature⁷. Only difference was the 3-times smaller magnetization strength of our material compared to the magnetic particles used in the previous report⁷. While the magnetic soft robot with data-driven structural and magnetic profile jumped to ~1.4 mm, 0.39 times its body length, the intuitive design adapted from the literature failed to produce any jumping at all. This comparison highlights the advancement provided by our data-driven design strategy to achieve improved functionality or movement performance.

In addition, we compared the shape complexity of our 2D shape-morphing demonstrations with previously reported results^{18,34-36}. This comparison revealed that our results showed up to 20 times more intricate profiles compared to previous literature (Fig. S8), despite the weaker magnetic actuation fields and magnetic strength of particles employed in our work.

Our data-driven design method demonstrates the capability to generate designs that utilize weaker materials while achieving similar behaviours to those produced by traditional methods. By simultaneously optimizing structural design and magnetization, our method compensates for the reduced magnetic strength of such materials through non-intuitive structural and magnetization configurations. The enhanced performance achieved using our framework could further enable other magnetic soft materials suffering in terms of performance (i.e., slow actuation, inferior mechanical properties) but are desirable in other aspects, including sustainability³⁷ and biocompatibility³⁸. For example, although biocompatible and biodegradable magnetic particles are heavily sought for medical applications³⁸, their weak magnetization strength and intuitive magnetic and morphological designs result in unsatisfactory performances for desired behaviors, which could be alleviated by the introduced data-driven approach.

11) The jumping experiment shown in Figure 3 is interesting. Is it possible to control the jumping direction through data-driven design?

Response: We thank the reviewer for their interest in the jumping experiment. Yes, it is indeed possible to control the jumping direction through data-driven design. We showed this capability in Fig. 4D, where a multi-material and 3D structured magnetic soft robot designed to maximize its directional jumping distance.

12) Design of multi-material and 3D magnetic soft millirobots is a creative idea. Zhang, J. et al. (doi.org/doi:10.1126/scirobotics.abf0112) assembled magnetic voxel units heterogeneous, demonstrating very complex locomotion such as rolling, rotation, and bistability, which can facilitate functions like drug transport and release. The current approach uses data-driven methods to assemble magnetic voxels, however, the results only show the jumping movements, which do not effectively illustrate the advantages of the current method. Could the author include additional applications to better demonstrate the benefits of the data-driven design?

Response: We thank the reviewer for highlighting the creativity of designing multi-material and 3D magnetic soft millirobots. Our main purpose of employing voxel-based multimaterial assembly was to highlight the fabrication, programming, and scale

agnostic nature of our data-driven design strategy and its applicability to stimuli responsive actuators fabricated using subtractive or additive manufacturing methods.
To clarify this point, we have included the statements below into the revised manuscript:

“In addition, scale of desired structures or robots, that could range from micrometer to centimeter, would require different fabrication techniques with critical implications for the design process. Therefore, the design strategies should be compatible with different fabrication techniques and capable of handling the enormous design space resulting from material selection, spatial stimuli response, and 3D structural complexity at varying scales. The data-driven strategy proposed here can also address the design challenge of soft materials with different fabrication methods, multi-material composition and 3D structures due to its generic and versatile framework (Fig. 4A). To demonstrate the compatibility of our approach in with other fabrication methods, we adapted voxel-based assembly²³ capable of fabricating 3D structures with multi-material compositions into our framework (Fig. S3). We created a material palette consisting of passive and magnetically responsive materials with soft and rigid variations (Fig. 4B, Fig. S14A and Table S4), expanding the design space by a power of 2.5 for any given structural workspace. The magnetization directions of the magnetic materials are discretized to six discrete primary cartesian axes directions to simplify fabrication process (Fig. 4B and Fig. S14B). The flexible nature of the proposed design strategy allows facile adaptation of the constraints for the parameter representation arising from the requirements of different fabrication methods, as well as material and structural complexity.”

Being said that, the benefits of our work are already demonstrated compared to the aforementioned report²³ in multiple points:

1. Unlike the demonstrations relying on the surface contact like rolling or rotation, as in the work of Zhang, J. et al.²³, we show a wide range of **jumping behaviors (Fig. 4) that is an inherently challenging locomotion mode for robots with soft bodies, requiring sudden release of energy to overcome the gravity.**
2. Our work achieves various locomotion modes with 10 mT compared to the much higher magnetic fields up to 530 mT utilized in the referenced work above, showcasing **our demonstrations are achieved much more efficiently with up to 53 times smaller magnetic fields.**
3. Further **extension to multi-stimuli responsive materials for the design of configurable robotic behaviors (Fig. S14).**

In addition to these points, mentioned locomotion modes of rolling and rotation before/after drug transport and release can be achieved by utilizing rotating magnetic fields as a control signal, as shown abundantly in the literature^{7,20,22,23}. Similar approaches, including utilization of net magnetization moment for rolling locomotion^{7,20,22,23} and visual feedback for programmed trajectory³⁹⁻⁴¹ could be easily applied in our untethered demonstrations as well. To showcase this capability, we **generated multimodal locomotion (jumping, rolling, traversing) on a pre-defined path with one of our 3D multimaterial robot designed for directional jumping (Fig. 4D) by employing dynamic magnetic fields, as shown below:**

In the revised manuscript:

“Additionally, incorporation of well-established control strategies, including the utilization of net magnetization moment for rolling locomotion^{7,20,22,23} and visual feedback for programmed trajectory³⁹⁻⁴¹, can enable multimodal locomotion modes for the untethered demonstrations (Fig. S16).”

In the revised supplementary notes:

Figure S16. Multimodal locomotion of the 3D multimaterial robot (Fig. 4D) on a pre-defined path. (A-B) The overview of multimodal locomotion of our 3D multimaterial robot (Fig. 4D) on a pre-defined path performing jumping (I), traversing (II), half rolling (III), and jumping (IV) to reach the desired end position (A), followed by half rolling (V), and rolling (VI) behaviors to get back to the starting position (B). The demonstrations are shown from top view. **(C-F)** The selected multimodal behaviors of jumping (C), traversing (D), half rolling (E) and backward rolling (F) achieved by the changes in the control signal (magnetic fields). The orange dashed

line is provided to highlight the change in the position of the robot in **(D)**. Scale bars, 4 mm. Actuation is performed by applying a uniform magnetic field (B) in the direction indicated by black arrows with 10 mT strength (C, E, F), or in the explicitly mentioned B vector (B_x , B_y , B_z) direction and strength (D).

Remarks on code availability:

I have read and successfully run the author's code on my computer.

Response: We thank the reviewer for testing and confirming the successful run of our code.

Reviewer #3 (Remarks to the Author):

The authors developed a data-driven design strategy for magnetic soft materials to achieve diverse 2D and 3D shape morphing and locomotion. The design framework is established by spatially programming the magnetization profile, morphology, and multi-material composition of magnetic soft materials, allowing them to achieve desired static or dynamic shape morphing behaviors under pre-defined external magnetic fields. Various demonstrations, including 2D/3D shape morphing of magnetic soft beams, jumping behaviors of magnetic soft millirobots, and controlled locomotion of multi-material 3D magnetic soft millirobots, showcase the robustness of the design strategy. The work is interesting, and the proposed design framework appears to be highly effective. I recommend this work for publication in Nature Communications after the following issues are addressed:

Response: We sincerely thank the reviewer for their thoughtful evaluation and positive feedback on our work. Below, we provide detailed responses to each of the issues raised, addressing them comprehensively and incorporating clarifications and revisions where necessary.

1. The authors used a mass-spring lattice model coupled with magnetic forces and torques as a simulation tool to support the data-driven design. However, no details on the implementation of the model are provided through reference provided. Supplementary examples should be provided regarding the static and dynamic models.

Response: We thank the reviewer for highlighting this point. Initially, we would like to clarify that the model is only dynamic, and the results obtained for shape-morphing demonstrations and the validation cases are the equilibrium states that are achieved during the dynamic simulation. Also, the code for our modified version of the simulation environment could be found in the shared link "[https://github.com/AlpKaracakol/data driven magnetic soft material design](https://github.com/AlpKaracakol/data_driven_magnetic_soft_material_design)".

The mass-spring lattice model is adapted from the work of S. Kriegman et al.⁶ and further details and additional implementations could be found in the original work of Hiller, J. et al.³. In our work, we modified the "Voxelyze" environment to integrate magnetic force and torque calculations. These modifications allow the simulation of magnetic materials interacting with either electromagnetic coil setups or permanent magnets, as used in our experimental demonstrations.

In our implementation, we simply calculate the magnetic forces and torques acting on each voxel, which are then included in the built-in total force and torque acting on each voxel. For experiments using only the electromagnetic coil setup (Fig. S2G), the magnetic field is uniform due to the Helmholtz coil configuration, and therefore, the magnetic forces are assumed to be negligible, leaving torques as the primary driver of deformation and motion. For the demonstrations using a permanent magnet setup (Fig. S2F), we calculated magnetic forces and torques by experimentally measuring magnetic field values at various distances from the magnet. Magnetic field gradients in z direction are calculated from these measurements, assuming negligible gradient in xy plane due to relatively uniform field at the operated distances. Based on the reviewer's comment, we have now added additional explanations regarding the simulation engine to the revised supplementary information (SI S2).

2. With an analytical model available, can the author discuss more about why stochastic optimization is chosen instead of gradient-based training of a neural network for the problems in the paper?

Response: We thank the reviewer for this question. The simulation model used in our framework is numerical rather than analytical, relying on Euler integration. This makes the computation of gradients with respect to design parameters challenging and resource-intensive. Stochastic optimization avoids the need for gradient computation and directly evaluates designs using the simulation model, making it more practical for our approach.

3. The optimization is less effective when predicting those with sharp turns in Fig. 2C and Fig. 2E. Is it possible to evaluate and decide the minimum radius of curvature you can have?

Response: We thank the reviewer for this insightful observation. Sharp corners in soft-bodied designs are inherently more challenging, which is precisely why we targeted shape-morphing demonstrations with sharp turns. These non-engineered, intricate shapes, derived from mathematical functions, were intentionally chosen to push the limits of our method, showcasing both its strengths and limitations. Furthermore, we intentionally refrained from engineering the target shapes or providing prior information to the framework, allowing the optimization to operate without bias.

While we agree with the reviewer that optimization results were less effective in Fig. 2C and 2E, we ascribe this to the complexity of overall desired shape, rather than the sharp turns. Although the square signal shape in Fig. 2B features sharp turns as well, the optimization algorithm was able to generate results with far less overall error and closer match at these sharp turns, indicating that the inability to capture the turns in Fig. 2C is not due to a limit in minimum radius of curvature. The minimum radius of curvature can be also improved by decreasing width of beam's cross-section, creating effective hinge joints that can generate large bending angles.

4. The desired static and dynamic shape morphing of magnetic soft materials is achieved by optimizing their magnetization and morphology. For shape morphing magnetic soft beams, the optimized designs often contain holes and imperfections. Is it possible to achieve these complex shape morphing using a smooth, perfect beam by combining the data-driven algorithm with theoretical beam models?

Response: We thank the reviewer for this thoughtful question. The holes and imperfections observed in the optimized designs are not artifacts but rather an intrinsic result of the optimization process, which explores all degrees of freedom, including structural voids, to achieve the desired shape morphing. These designs highlight the framework's ability to fully leverage the available design space for optimizing performance.

The data-driven algorithm operates within a voxelized design space, where the inclusion of voids and structural heterogeneities provides greater flexibility in achieving complex deformations. Such features enable more localized control of bending, twisting, and shape transitions, which may be challenging to replicate with smooth, continuous beams. By contrast, a continuous beam introduces constraints on the design space, reducing the degrees of freedom available for optimization. This

limitation may restrict the framework's ability to achieve intricate or non-intuitive shape morphing behaviors.

Furthermore, existing theoretical beam models²⁵⁻³⁰ are inherently limited in their scope. They typically assume simplified structural designs, 2D deformations, or simplified magnetization profiles (1D or only cartesian axis). Integrating such models with the data-driven algorithm would impose additional constraints on the optimization process, potentially diminishing the achievable shape-morphing capabilities.

5. A marginal change in the beam structure and magnetization can lead to dramatic configuration differences. Is this related to the snap-through of slender magnetic beam structures? For a given beam structure like the one in Fig. S1A, would applying the magnetic field at different speeds or different paths with the slight out-of-plane component cause the beam to deform into a different configuration in Fig. S1B?

Response: We thank the reviewer for this thought-provoking query. Based on the reviewer's comments, we have performed extensive simulations and observed that the dramatic change in shape-morphing with respect to marginal changes in magnetization and structure is not due to the snap-through behavior of the magnetic beam. When the field was gradually increased, instead of sudden application, the resulting deformations showed negligible changes (Fig. S8). On the other hand, a magnetic field following a rotating path from -z to +z direction resulted in dramatic shape-morphing configurations. The slight out-of-plane components on top of the gradually increasing or rotating magnetic fields mostly affected the alignment in x-y plane (Fig. S8). However, in specific cases, such as the 2nd column 4th row, out of plane components can also affect the resulting configuration drastically. These results show that resulting equilibrium state for a given design depends on the induced magnetic field input sequences to the system.

To further investigate the sensitivity of shape-morphing based on changes of magnetization direction within specific segments, we generated a heat-map of shape-morphing change ratio by varying the magnetization direction in increments of 15° from -180° to 180°, in all 20 different segments (Fig. S7). The resulting heat-map shows the inherently non-linear design space with numerous local minima both with different segment numbers and magnetization directions. These results shows that there are a wide range of equilibrium states for a given design and the resulting equilibrium state depends on the initial state and the induced magnetic field input sequences to the system.

Based on the reviewer's comment, we have included the following statements and the figures in the revised manuscript and supplementary notes, respectively:

"We further validated the non-intuitive nature of the coupled design space by introducing marginal changes in magnetization and morphology of the sinusoidal shape-morphing beam, as well as the external magnetic field, which resulted in dramatic shifts from the desired shape in the simulation environment (Figs. S6-S8). These sensitivity analysis results show that the design space is inherently non-linear with numerous local minima and there are a wide range of equilibrium states for a given design, which depends on the initial design state (Figs. S7) and the induced magnetic field input sequences (Fig. S8) to the system."

Figure S8. Sensitivity analysis for the effect of applied magnetic field on shape-morphing of magnetic soft beams. (A) Simulation results are presented for the original design (1st column), a disturbed magnetization profile (2nd column), and a disturbed

morphology (3rd column) under a range of external magnetic fields (rows). In the 2nd column, the magnetization of the 11th segment is changed 30°, corresponding to a 0.4% change in the overall magnetic profile, and in the 3rd column, the morphology of the design is changed by cutting a single voxel line laterally corresponding to a 1.5% change in overall morphology (Fig. 1B). **(B)** The rows represent different magnetic field configurations: the 1st row shows the original magnetic field with a sudden applied 30 mT (B^0), the 2nd row shows a gradually increasing magnetic field (B^I), the 3rd row includes a gradually increased magnetic field with a 0.3 mT out-of-plane in y direction component (B^I+B^{III}), 4th row includes gradually increased magnetic field combined with a 1.5 mT out-of-plane in y direction component (B^I+B^{IV}), 5th row shows a rotating magnetic field from -z to +z direction (B^{II}), 6th row combines the rotating magnetic field from -z to +z direction with a 0.3 mT out-of-plane in y direction component ($B^{II}+B^{III}$), 7th row combines a rotating magnetic field from -z to +z direction with a 1.5 mT out-of-plane in y direction component ($B^{II}+B^{IV}$). Color bars indicate the average strain energy density.

Figure S7. Sensitivity heatmap and selected examples for the effect of magnetization on shape-morphing of magnetic soft material. (A) The sensitivity heatmap showing the shape-morphing change ratio (%), for the demonstration in Fig. 2A, based on the relative change of the magnetization direction (θ , y-axis) at different segment numbers (x-axis). The heatmap and illustrative cases highlight the non-linear and non-intuitive behavior of magnetic soft materials. The selected examples are marked with orange boxes, and their corresponding

shape-morphing results are linked via orange lines. Shape-morphing change ratio is calculated as the average displacement error, with respect to the original design, normalized to the body length. Color bars represent the shape-morphing change ratio (%) and average strain energy density (J/m^3). **(B)** The shape-morphing of the original design is shown for comparison, along with the description of the relative rotation change (θ), and the respective segment numbers of the given original design.

6. The author showed the marginal effect that a small change in beam length can cause drastic deformation differences. For the 3D dynamic robot, would the manual assembly of each voxel lead to fabrication errors and behaviors different from targeted ones?

Response: We thank the reviewer for this comment. For the voxel-assembly method we acknowledge that the fabrication involves manual handling of the samples at every step of the fabrication (FIG. S3), making it highly prone to fabrication artifacts. To minimize potential handling and fabrication errors, we have employed voxels with a side length of 2 mm, resulting in negligible imperfections on the assembled structure. Furthermore, the voxel-assembly demonstrations are not fixed-end but rather have multiple contact points on the surface, which further eliminates the potentially drastic effect of the artifacts, in comparison to small scale and fixed-end beam demonstrations. Overall, we showcased 3D dynamic robot demonstrations primarily to highlight fabrication, programming, and scale agnostic nature of our approach.

7. The author's group has already demonstrated various soft robotics with incredible moving capabilities by rolling or tumbling beam structures. Fig.4 demonstrates multiple 3D robot designs with much more complicated structures but weaker moving performance. How would the authors justify the merits of using the current design framework to design these 3D and multilateral robots?

Response: We thank the reviewer for the feedback. We agree that our group demonstrated various soft robots with locomotion capabilities by rolling or tumbling structures^{7,18}. Our main purpose of employing voxel-based 3D multimaterial assembly was to highlight the fabrication, programming, and scale agnostic nature of our data-driven design strategy and its applicability to stimuli responsive actuators fabricated using subtractive or additive manufacturing methods. To clarify this point, we have included the statements below into the revised manuscript:

“In addition, scale of desired structures or robots, that could range from micrometer to centimeter, would require different fabrication techniques with critical implications for the design process. Therefore, the design strategies should be compatible with different fabrication techniques and capable of handling the enormous design space resulting from material selection, spatial stimuli response, and 3D structural complexity at varying scales. The data-driven strategy proposed here can also address the design challenge of soft materials with different fabrication methods, multi-material composition and 3D structures due to its generic and versatile framework (Fig. 4A). To demonstrate the compatibility of our approach in with other fabrication methods, we adapted voxel-based assembly²³ capable of fabricating 3D structures with multi-material compositions into our framework (Fig. S3). We created a material palette consisting of passive and magnetically responsive materials with soft and rigid variations (Fig. 4B, Fig. S11A and Table S4), expanding the design space by a power of 2.5 for any given structural workspace. The magnetization directions of the magnetic

materials are discretized to six discrete primary cartesian axes directions to simplify fabrication process (Fig. 4B and Fig. S11B). The flexible nature of the proposed design strategy allows facile adaptation of the constraints for the parameter representation arising from the requirements of different fabrication methods, as well as material and structural complexity.”

In addition, mentioned locomotion modes of rolling and tumbling can be achieved by utilizing rotating or repeating magnetic fields as a control signal, as shown abundantly in the literature^{7,20,22,23}. Similar approaches, including utilization of net magnetization moment for rolling locomotion and visual feedback for programmed trajectory³⁹⁻⁴¹ could be easily applied in our untethered demonstrations as well. To showcase this capability, we generated multimodal locomotion (jumping, rolling, traversing) on a pre-defined path with one of our 3D multimaterial robot designed for directional jumping (Fig. 4D) by employing dynamic magnetic fields, as shown below:

In the revised manuscript:

“Additionally, incorporation of well-established control strategies, including the utilization of net magnetization moment for rolling locomotion^{7,20,22,23} and visual feedback for programmed trajectory³⁹⁻⁴¹, can enable multimodal locomotion modes for the untethered demonstrations (Fig. S16).”

In the revised supplementary notes:

Figure S16. Multimodal locomotion of the 3D multimaterial robot (Fig. 4D) on a pre-defined path. (A-B) The overview of multimodal locomotion of our 3D multimaterial robot (Fig. 4D) on a pre-defined path performing jumping (I), traversing (II), half rolling (III), and jumping (IV) to reach the desired end position (A), followed by half rolling (V), and rolling (VI) behaviors to get back to the starting position (B). The demonstrations are shown from top view. **(C-F)** The selected multimodal behaviors of jumping (C), traversing (D), half rolling (E) and backward rolling (F) achieved by the changes in the control signal (magnetic fields). The orange dashed line is provided to highlight the change in the position of the robot in (D). Scale bars, 4 mm. Actuation is performed by applying a uniform magnetic field (B) in the direction indicated by black arrows with 10 mT strength (C, E, F), or in the explicitly mentioned B vector (B_x , B_y , B_z) direction and strength (D).

8. There is a typo in the manuscript. In the third paragraph of the Section “Jumping behavior for magnetic soft millirobots”, the figure citation “Fig. 10B” should be “Fig. S10B”. The readability of SI can be improved.

Response: We thank the reviewer for this comment. We corrected it.

References

- 1 Kocaman Kabil, F. & Oral, A. Y. Influence of the pore size on optical and mechanical properties of ecoflex sponges. *Materials Research Express* **11**, 035305 (2024). <https://doi.org/10.1088/2053-1591/ad2a87>
- 2 Magnequench, <<https://mgitechnology.com/wp-content/uploads/2017/09/mgp-10-85-20180-070.pdf#page=1.00>> (2020).
- 3 Hiller, J. & Lipson, H. Dynamic Simulation of Soft Multimaterial 3D-Printed Objects. *Soft Robotics* **1**, 88-101 (2014). <https://doi.org/10.1089/soro.2013.0010>
- 4 Cheney, N., Bongard, J. & Lipson, H. Evolving Soft Robots in Tight Spaces. *Proceedings of the 2015 Annual Conference on Genetic and Evolutionary Computation*, 935–942 (2015). <https://doi.org/10.1145/2739480.2754662>
- 5 Cheney, N., MacCurdy, R., Clune, J. & Lipson, H. Unshackling evolution: evolving soft robots with multiple materials and a powerful generative encoding. *SIGEVOlution* **7**, 11–23 (2014). <https://doi.org/10.1145/2661735.2661737>
- 6 Kriegman, S., Blackiston, D., Levin, M. & Bongard, J. A scalable pipeline for designing reconfigurable organisms. *Proceedings of the National Academy of Sciences* **117**, 1853-1859 (2020). <https://doi.org/10.1073/pnas.1910837117>
- 7 Hu, W., Lum, G. Z., Mastrangeli, M. & Sitti, M. Small-scale soft-bodied robot with multimodal locomotion. *Nature* **554**, 81-85 (2018). <https://doi.org/10.1038/nature25443>
- 8 Li, M., Pal, A., Aghakhani, A., Pena-Francesch, A. & Sitti, M. Soft actuators for real-world applications. *Nature Reviews Materials* **7**, 235-249 (2022). <https://doi.org/10.1038/s41578-021-00389-7>
- 9 Xia, X., Spadaccini, C. M. & Greer, J. R. Responsive materials architected in space and time. *Nature Reviews Materials* **7**, 683-701 (2022). <https://doi.org/10.1038/s41578-022-00450-z>
- 10 Sitti, M. Physical intelligence as a new paradigm. *Extreme Mechanics Letters* **46**, 101340 (2021). <https://doi.org/10.1016/j.eml.2021.101340>
- 11 Kim, Y. & Zhao, X. Magnetic Soft Materials and Robots. *Chemical Reviews* **122**, 5317-5364 (2022). <https://doi.org/10.1021/acs.chemrev.1c00481>
- 12 Park, S.-J. *et al.* Phototactic guidance of a tissue-engineered soft-robotic ray. *Science* **353**, 158-162 (2016). <https://doi.org/doi:10.1126/science.aaf4292>
- 13 Pikul, J. H. *et al.* Stretchable surfaces with programmable 3D texture morphing for synthetic camouflaging skins. *Science* **358**, 210-214 (2017). <https://doi.org/doi:10.1126/science.aan5627>
- 14 Boley, J. W. *et al.* Shape-shifting structured lattices via multimaterial 4D printing. *Proceedings of the National Academy of Sciences* **116**, 20856-20862 (2019). <https://doi.org/doi:10.1073/pnas.1908806116>
- 15 Flavin, M. T. *et al.* Bioelastic state recovery for haptic sensory substitution. *Nature* **635**, 345-352 (2024). <https://doi.org/10.1038/s41586-024-08155-9>
- 16 Kim, Y. *et al.* Telerobotic neurovascular interventions with magnetic manipulation. *Science Robotics* **7**, eabg9907 (2022). <https://doi.org/doi:10.1126/scirobotics.abg9907>
- 17 Rumley, E. H. *et al.* Biodegradable electrohydraulic actuators for sustainable soft robots. *Science Advances* **9**, eadf5551 (2023). <https://doi.org/doi:10.1126/sciadv.adf5551>

- 18 Lum, G. Z. *et al.* Shape-programmable magnetic soft matter. *Proceedings of the National Academy of Sciences* **113**, E6007-E6015 (2016). <https://doi.org/doi:10.1073/pnas.1608193113>
- 19 Cui, J. *et al.* Nanomagnetic encoding of shape-morphing micromachines. *Nature* **575**, 164-168 (2019). <https://doi.org/10.1038/s41586-019-1713-2>
- 20 Kim, Y., Yuk, H., Zhao, R., Chester, S. A. & Zhao, X. Printing ferromagnetic domains for untethered fast-transforming soft materials. *Nature* **558**, 274-279 (2018). <https://doi.org/10.1038/s41586-018-0185-0>
- 21 Alapan, Y., Karacakol, A. C., Guzelhan, S. N., Isik, I. & Sitti, M. Reprogrammable shape morphing of magnetic soft machines. *Science Advances* **6**, eabc6414 (2020). <https://doi.org/doi:10.1126/sciadv.abc6414>
- 22 Xu, T., Zhang, J., Salehizadeh, M., Onaizah, O. & Diller, E. Millimeter-scale flexible robots with programmable three-dimensional magnetization and motions. *Science Robotics* **4**, eaav4494 (2019). <https://doi.org/doi:10.1126/scirobotics.aav4494>
- 23 Zhang, J. *et al.* Voxellated three-dimensional miniature magnetic soft machines via multimaterial heterogeneous assembly. *Science Robotics* **6**, eabf0112 (2021). <https://doi.org/doi:10.1126/scirobotics.abf0112>
- 24 Abbott, J. J., Diller, E. & Petruska, A. J. Magnetic Methods in Robotics. *Annual Review of Control, Robotics, and Autonomous Systems* **3**, 57-90 (2020). <https://doi.org/10.1146/annurev-control-081219-082713>
- 25 Ruike, Z., Yoonho, K., Shawn, A. C., Pradeep, S. & Xuanhe, Z. Mechanics of hard-magnetic soft materials. *Journal of the Mechanics and Physics of Solids* **124**, 244-263 (2019). <https://doi.org/10.1016/j.jmps.2018.10.008>
- 26 Tomohiko, G. S., Matteo, P. & Pedro, M. R. A Kirchhoff-like theory for hard magnetic rods under geometrically nonlinear deformation in three dimensions. *Journal of the Mechanics and Physics of Solids* **160**, 104739 (2022). <https://doi.org/https://doi.org/10.1016/j.jmps.2021.104739>
- 27 Wei, C., Zhi, Y. & Lin, W. On mechanics of functionally graded hard-magnetic soft beams. *International Journal of Engineering Science* **157**, 103391 (2020). <https://doi.org/https://doi.org/10.1016/j.ijengsci.2020.103391>
- 28 Wang, J., Wang, D., Dong, L., Zhang, M. & Gu, G. Analytical Modeling and Inverse Design of Centimeter-Scale Hard-Magnetic Soft Robots. *IEEE Transactions on Automation Science and Engineering* **21**, 5558-5569 (2024). <https://doi.org/10.1109/TASE.2023.3313395>
- 29 Yan, D., Abbasi, A. & Reis, P. M. A comprehensive framework for hard-magnetic beams: Reduced-order theory, 3D simulations, and experiments. *International Journal of Solids and Structures* **257**, 111319 (2022). <https://doi.org/https://doi.org/10.1016/j.ijsolstr.2021.111319>
- 30 Dreyfus, R., Boehler, Q. & Nelson, B. J. A Simulation Framework for Magnetic Continuum Robots. *IEEE Robotics and Automation Letters* **7**, 8370-8376 (2022). <https://doi.org/10.1109/LRA.2022.3187249>
- 31 MacDonald, R. E. & Beck, J. W. Magneto-Optical Recording. *Journal of Applied Physics* **40**, 1429-1435 (1969). <https://doi.org/10.1063/1.1657706>
- 32 Kryder, M. H. Magneto-optic recording technology (invited). *Journal of Applied Physics* **57**, 3913-3918 (1985). <https://doi.org/10.1063/1.334915>
- 33 Challener, W. A. *et al.* Heat-assisted magnetic recording by a near-field transducer with efficient optical energy transfer. *Nature Photonics* **3**, 220-224 (2009). <https://doi.org/10.1038/nphoton.2009.26>

- 34 Wu, S. *et al.* Evolutionary Algorithm-Guided Voxel-Encoding Printing of Functional Hard-Magnetic Soft Active Materials. *Advanced Intelligent Systems* **2**, 2000060 (2020). <https://doi.org/10.1002/aisy.202000060>
- 35 Wang, L. *et al.* Evolutionary design of magnetic soft continuum robots. *Proceedings of the National Academy of Sciences* **118**, e2021922118 (2021). <https://doi.org/doi:10.1073/pnas.2021922118>
- 36 Lloyd, P. *et al.* A Magnetically-Actuated Coiling Soft Robot With Variable Stiffness. *IEEE Robotics and Automation Letters* **8**, 3262-3269 (2023). <https://doi.org/10.1109/LRA.2023.3264770>
- 37 Patel, D. K., Zhong, K., Xu, H., Islam, M. F. & Yao, L. Sustainable Morphing Matter: Design and Engineering Practices. *Advanced Materials Technologies* **8**, 2300678 (2023). <https://doi.org/10.1002/admt.202300678>
- 38 Goudu, S. R. *et al.* Biodegradable Untethered Magnetic Hydrogel Milli-Grippers. *Advanced Functional Materials* **30**, 2004975 (2020). <https://doi.org/10.1002/adfm.202004975>
- 39 Yao, J. *et al.* Adaptive Actuation of Magnetic Soft Robots Using Deep Reinforcement Learning. *Advanced Intelligent Systems* **5**, 2200339 (2023). <https://doi.org/10.1002/aisy.202200339>
- 40 Demir, S. O. *et al.* Task space adaptation via the learning of gait controllers of magnetic soft millirobots. *The International journal of robotics research* **40**, 1331-1351 (2021).
- 41 Culha, U., Demir, S. O., Trimpe, S. & Sitti, M. Learning of sub-optimal gait controllers for magnetic walking soft millirobots. *Robotics science and systems: online proceedings* **2020** (2020).